



# The influence of decadal oscillations on the oxygen and nutrient trends in the Pacific Ocean

Lothar Stramma[1], Sunke Schmidtko[1], Steven J. Bograd[2], Tsuneo Ono[3], Tetjana Ross[4], Daisuke Sasano[5], and Frank A. Whitney[4]

5 [1]GEOMAR Helmholtz Centre for Ocean Research Kiel, Düsternbrooker Weg 20, 24105 Kiel, Germany

[2]Environmental Research Division, Southwest Fisheries Science Center, NOAA, Monterey, California, USA

[3]National Research Institute for Far Sea Fisheries, Fisheries Research and Education Agency, 2-12-4 Fukuura, Kanazawa-Ku, Yokohama 236-8648, Japan

10 [4]Institute of Ocean Sciences, Fisheries and Oceans Canada, PO Box 6000, Sidney BC V8L 4B2, Canada

[5]Global Environment and Marine Department, Japan Meteorological Agency, Tokyo, Japan

*Correspondence to*: Lothar Stramma (lstramma@geomar.de)



**Abstract.** A strong oxygen deficient layer is located in the upper layer of the tropical Pacific Ocean and at deeper depths in the North Pacific. Processes related to climate change (upper ocean warming, reduced ventilation) are expected to change ocean oxygen and nutrient inventories. In most ocean basins, a decrease in oxygen ('deoxygenation') and an increase of nutrients has been observed in subsurface layers. Deoxygenation trends are not linear and there could be other influences on oxygen and nutrient trends and variability. Here oxygen and nutrient time series since 1950 in the Pacific Ocean were investigated at 50 to 300 m depth, as this layer provides critical pelagic habitat for biological communities. In addition to trends related to ocean warming the oxygen and nutrient trends show a strong influence of the Pacific Decadal Oscillation (PDO) in the tropical and the eastern Pacific, and the North Pacific Gyre Oscillation (NPGO) especially in the North Pacific. In the Oyashio Region the PDO, the NPGO, the North Pacific Index (NPI) and a 18.6 year nodal tidal cycle overlay the long-term trend. In most regions oxygen increases and nutrients decrease in the 50 to 300 m layer during the negative PDO phase, with opposite trends during the positive PDO phase. The PDO index encapsulates the major mode of surface temperature variability in the Pacific and oxygen and nutrients trends throughout the basin can be described in the context of the PDO phases. An influence of the subtropical-tropical cell in the tropical Pacific cannot be proven with the available data. El Niño and La Niña years often influence the oxygen and nutrient distribution during the event in the eastern tropical Pacific, but do not have a multi-year influence on the trends.

## 1 Introduction

Oxygen and nutrient distribution are key parameters controlling marine ecosystems. How oxygen and nutrient concentrations vary and co-vary ultimately controls biogeochemical cycles. Globally, oxygen has been estimated to have decreased in the ocean by 2% during the past five decades, likely caused by climate change related temperature increases, with the largest oxygen decrease in the North and equatorial Pacific (Schmidtko et al., 2017). Increasing sea surface temperature reduces the solubility of oxygen in sea water and increases stratification which leads to less convection of oxygen rich water to subsurface layers. The global mean surface temperature (GISTemp) anomaly (e.g. Hansen et al., 2010) shows an increase of about 0.3°C from early 1900 to 1950, stagnant values between 1950 to 1976 and



an increase of about 0.8°C from 1976 to 2015 (Fig. 1). Results from the UVic ECSM (University of Victoria Earth System Climate Model) model indicate that ocean oxygenation varies inversely with low-latitude surface wind stress (Ridder and England 2014).

In the tropical eastern Pacific, two subsurface low-oxygen zones exist north and south of the equator,
with a pronounced minimum in oxygen at ~100 to 500 m depth, and are referred to as oxygen minimum zones (OMZ's) or oxygen deficient zone (ODZ's). These OMZ's are suboxic (oxygen concentrations below ~4.5-10.0 μmol kg$^{-1}$; e.g. Karstensen et al., 2008; Stramma et al., 2008). In suboxic regions, nitrate and nitrite become involved in respiration processes such as denitrification or anammox (e.g. Kalvelage et al., 2013). Decreasing and, in few areas of the Pacific, increasing oxygen content in the
OMZ layer over the last 50 years has been described (e.g. Stramma et al., 2010). In the subarctic North Pacific surface nutrient concentration decreased during 1975 to 2005, and is strongly correlated with a multidecadal increasing trend of sea surface temperature (SST) (Ono et al., 2008). Below the surface, however, oxygen decreased and nutrients increased in the subarctic Pacific pycnocline from the mid-1980s to around 2010 (Whitney et al., 2013). Nutrients would be expected to vary inversely with
oxygen, if the dominant process was the remineralization of marine detritus (Whitney et al., 2013).

Climate modes influence oxygen and nutrient distributions. Because of the influence of SST on the solubility of oxygen, the most prominent control on oxygen changes in the Pacific might be exerted by the Pacific Decadal Oscillation (PDO). A shoaling thermocline, such as occurs in the eastern Pacific during La Niña or cool (negative) PDO state, enhances nutrient supply and organic matter export while
simultaneously increasing the fraction of that organic matter that is respired in the low-oxygen water of the uplifted thermocline. The opposite occurs during El Niño or a warm (positive) PDO state; a deeper thermocline reduces both export and respiration in low-oxygen water, allowing the hypoxic water volume to shrink (Deutsch et al., 2011; Fig. S7). Previous syntheses of tropical and North Pacific physical, biological and chemical conditions during warm and cold PDO regimes have shown the far-
reaching influence of the PDO on the Pacific Ocean, for example in controlling the out-of-phase relationship between sardine and anchovy populations in the eastern Pacific (Chavez et al., 2003). In a pattern similar to the PDO SST during the warm regime, lower nutrients are shown for the eastern

equatorial and near-coastal tropical Pacific and higher nutrients in the northern Pacific, and vice-versa for the cold regime (Chavez et al., 2003; their Fig. 3). A similar relationship to the PDO was also observed for surface (5-10 m) nutrient concentrations in the North Pacific (north of 10°N, Yasunaka et al. 2016). Model simulations for the eastern Pacific Ocean for typical PDO positive conditions show a

volume expansion of the suboxic regions by 7% in 50 years due to a slow-down of the large scale circulation related to the decrease of the intensity of the trade winds (Duteil et al., 2018). Other climate modes that could influence the oxygen and nutrient trends are the North Pacific Gyre Oscillation (NPGO; Di Lorenzo et al., 2008), the Pacific subtropical-tropical cell (STC; e.g. Duteil et al., 2014), the North Pacific Index (NPI) and El Niño-Southern Oscillation (ENSO) events.

Here we use publicly available oxygen and nutrient data augmented with recent ship data to investigate the influence of decadal climate oscillations on the oxygen and nutrient variability in the tropical, eastern and northern Pacific Ocean where long time series are available. For the negative PDO phase we use data between 1950 and 1976. As there are large data gaps in the 1990s and early 2000s we use data since 1977 for the warm PDO despite the variable PDO conditions after 1998. In addition, a possible

influence of the STC, of ENSO and of a 18.6 year oscillation (Royer 1993) in the North Pacific is investigated.

## 2 Climate signals and data sets

### 2.1    Climate signals

Several climate signals are investigated here with regard to a possible influence on the oxygen and

nutrient distribution and some basic details are listed here.

The PDO is the leading empirical orthogonal function (EOF) of monthly SST over the North Pacific between 20°N and 70°N, after removing the global mean SST anomaly and its associated principal component (PC) time series (Mantua et al., 1997; Deser et al., 2010; Newman et al., 2016). The temperature pattern and the PDO time series (Supplement Fig. S1) show that the PDO was negative

during 1944 to 1976 with a stagnant temperature period, and warming temperatures since 1977 with PDO positive during 1977 to 1998 and PDO variable after 1998. At the time of transition from negative



to positive PDO a climate shift in the eastern and central North Pacific Ocean occurred in 1976-77, which was caused by unique atmospheric anomalies acting over several months before the 1976-77 winter (Miller et al., 1994). Despite a positive PDO index from 2002 to 2006 (Fig. 1), the period 1998 to 2013 is dominated by negative seasonal mean PDO indices and is typically considered as a cool

(negative PDO) phase (Trenberth 2015).

The North Pacific Gyre Oscillation (NPGO) index tracks the changes in strength of the central and eastern branches of the North Pacific gyres and of the Kuroshio-Oyashio Extension. Like the PDO, it is a mode of decadal climate variability and it is defined as the second dominant mode of variability in sea surface height anomaly in the northeast Pacific over the region 180-110°W, 25-62°N (Di Lorenzo et al.,

2008). Subsurface nutrient variability in the Gulf of Alaska, Line P in the eastern North Pacific at about 50°N and the California Current System have been shown to be correlated with the NPGO (Di Lorenzo et al., 2009).

A decadal decline of the Pacific subtropical cells (STC) from the 1950s into the 1990s was computed between 9°N and 9°S with a decreased equatorward convergence of 12 Sv (McPhaden and Zhang

2002). This was followed by an increase of STC convergence from the mid-1990s to about 2000 (McPhaden and Zhang 2004). The transport changes observed in the Pacific STC (e.g. Schott et al., 2008), with a decrease of 30% from the 1960s to the 1990s, resulted in reduced transports of oxygen and phosphate to the tropics in model simulations (Duteil et al., 2014). In model computations the decadal variability of STC in the Northern Hemisphere was found to be strongly associated with the

PDO while in the south the STC only passively responds to the PDO (Hong et al., 2014).

In the North Pacific, the NPI (North Pacific Index; anomaly of the sea surface level pressure in the wintertime of the North Pacific (160°E-140°W, 30°N-65°N; Minobe, 2000) was introduced. In observations in the Northwest Pacific a bi-decadal oscillation related to minima in 1962 and 1983 and maxima in 1971 and 1991 (Watanabe et al., 2003, their Fig. 2) has been described.

The ENSO cycle of alternating warm El Niño and cold La Niña events is the climate system's dominant year-to year signal. ENSO originates in the tropical Pacific through interaction between the ocean and the atmosphere, but its environmental and socioeconomic impacts are felt worldwide (McPhaden et al.,



2006). Three month running mean sea surface temperature anomalies (ERSST.v5 SST anomalies) in the Niño 3.4 region (equatorial Pacific: 5°N to 5°S, 120°W to 170°W) of at least +0.5°C and lasting for at least 5 consecutive months are defined as El Niño events and 5 months of at least -0.5°C are defined as La Niña events (http://origin.cpc.ncep.noaa.gov/products/analysis_monitoring/ensostuff/ONI_v5.php).

In the eastern Pacific, ENSO variability is most pronounced along the Equator and coastal Ecuador, Peru and Baja California (Wang and Fiedler, 2006). Coastal warming during El Niño is caused by downwelling Kelvin waves generated by mid-Pacific westerly wind anomalies that deepen the eastern thermocline, nutricline and oxycline and allow warming to occur (Kessler, 2006). Consequently, during El Niño events the upwelled water off the coast is warmer, more oxygen replete and less nutrient-rich

while during La Niña events the water is colder, oxygen poor and nutrient-rich (e.g., Graco et al., 2017).

## 2.2 Data Sources by Regions

The main hydrographic dataset is similar to the one used and described in Schmidtko et al. (2017), relying on Hydrobase and World Ocean Database bottle data for nutrient data. Quality control and handling is described in Schmidtko et al. (2017) for oxygen and used here similarly for nutrients. The

only divergence to the described procedure was to ensure all data were in µmol kg$^{-1}$ and not requiring the discarding already sparse data: bottle data with missing temperature and/or salinity were assigned the temporal and spatial interpolated temperature and salinity derived from MIMOC (Schmidtko et al., 2013). This may lead to minor errors in density on the order of ~0.5 kg m$^{-3}$, thus about 0.05%. Similarly, a nutrient file was compiled (status 25 February 2019). This compilation of hydrographic,

oxygen and nutrient data are referred to in the following as 'hydrodata'. The final hydrodata set was used to extract data for our regions of interest.

As nutrient data are sparse in many regions of the Pacific, areas were selected due to their better temporal data coverage (most regular sampling over the longest time period). Three regions in the equatorial Pacific were selected which can be compared with earlier observations. In the equatorial

region mainly hydrodata were used. The region at 5°N-5°S, 165°-175°W (area E in Stramma et al., 2008; shown in Fig. 2a) which had hydrodata until 2009, was supplemented with data from a *R/V Investigator* cruise at 170°W from June 2016. The region 5°N-5°S, 105°-115°W (area D in Stramma et

al. 2008; shown in Fig. 2a), which had hydrodata until 2008, was supplemented with data from a RV *Ron Brown* cruise at 110°W in December 2016. The area 2°S-5°S, 84°-87°W (near the Galapagos Islands, marked in Fig. 2a as area G) had been used by Czeschel et al. (2015) to investigate nutrient trends and is supplemented here with recent cruises until 2017.

Off California, intense repeated measurements have been made since 1949 within the California Cooperative Oceanic Fisheries Investigations (CalCOFI) project and data from the bottle data set (downloaded from http://www.calcofi.org/ccdata.html, status 13 August 2018; data period March 1949 to November 2017, however with sparse nutrient data prior to 1984) were used to investigate the oxygen and nutrient changes within the California Current. We use the CalCOFI 1x1° data subset at 34-

35°N, 121-122°W without additional hydrodata and no additional historic nutrient data. This region is located near the center of the CalCOFI station grid and we referred to it as CalCOFIc (shown in Fig. 2a).

Station P, located at 50°N, 145°W in the North Pacific, is an ocean measurement site established in 1943 which was manned continuously until 1981, and then on average 3 times a year since. The data set

from the Institute of Ocean Sciences, Sidney BC, Canada covers (status September 2018) the time period May 1956 to August 2017 with data collected on research cruises. The Station P data were supplemented here by hydrodata in the surrounding region 48°-52°N, 143°-147°W, which we refer to as area P (shown in Fig. 2a) in the following.

The Hawaii Ocean Time-series for the region north of Hawaii was taken from hydrodata in the region

22-25°N, 156-159°W and downloaded bottle data for the Aloha station at 22°45'N, 158°W (http://hahana.soest.hawaii.edu/hot/hot-dogs/bextraction.html status 15 January 2019, time period covered October 1988 to December 2017). This data set is called Aloha region (shown in Fig. 2a) in the following. Different to the other regions this station is located in the subtropics with high oxygen content and low nutrient concentrations in the subsurface layer (Fig. 2). As there were continuous

measurements since the 1980s we included this region in our investigation.

The measurements in the Oyashio region east of Japan (39°-42°N, 144-149°E; shown in Fig. 2a)) are from hydrodata, augmented with updated data collections used in Whitney et al. (2013). Hydrographic





and biogeochemical measurements have typically been made in this region in every season since 1954 (used in Sasano et al., 2018) and were included here up to 2017.

In the eastern tropical Pacific (2°S-5°S, 84°-87°W; area G) and off Peru (7-12°S, 78-83°W; called Peru region in the following) the hydrodata could be extended with some RV *Meteo*r cruises carried out across the equator and near the Peruvian coast, with two cruise legs from December 2008 to February 2009 (M77/3 and M77/4; Czeschel et al., 2011), in November 2012 (M90; Stramma et al., 2013) at 85°50'W and in June 2017 (M138). Sections across the Peruvian shelf between 9°S and 16°S were made during RV *Meteor* cruise M91 in December 2012 (Czeschel et al., 2015) and M135 in March 2017. In October 2015 an RV *Sonne* cruise from Guayaquil, Ecuador to Antofagasta, Chile, was carried out with sections at the equator and off the shelf of Peru (Stramma et al., 2016). Although these measurements were made during one of the strongest El Niño events since the 1950s, the October 2015 measurements are used for the trend computations to estimate the influence of the El Niño event. In the figures the El Niño events of 1982/83, 1997/98 and 2015/16 are defined as very strong El Niño events and the strong El Niño events (1957/58, 1965/66, 1972/73, 1987/88 and 1991/92) are marked by circles and the strong La Niña events (1973/74, 1975/76, 1988/89, 1998/99, 1999/2000, 2007/08 and 2010/11) are marked by squares in these years.

## 2.3    Data Processing

On the recent cruises the CTD oxygen sensors were calibrated with oxygen measurements obtained from discrete samples from the rosette applying the classical Winkler titration method, using a non-electronic titration stand (Winkler, 1888; Hansen, 1999). The root mean square uncertainty of the CTD oxygen sensor calibration was on the order of $\pm 1.0$ µmol kg$^{-1}$.

Nutrients nitrite ($NO_2^-$), nitrate ($NO_3^-$), phosphate ($PO_4^{3-}$) and silicic acid ($Si(OH))_4$ referred to as silicate hereafter) on the recent cruises were measured on-board with a QuAAtro auto-analyzer (Seal Analytical). For recent autoanalyzer measurements precisions are 0.01 µmol kg$^{-1}$ for phosphate, 0.1 µmol kg$^{-1}$ for nitrate, and 0.5 µmol kg$^{-1}$ for silicate and 0.02 mL L$^{-1}$ (~ 0.9 µmol kg$^{-1}$) for oxygen from Winkler titration (Bograd et al., 2015). For older uncorrected nutrient data, offsets are estimated to be 3.5% for nitrate, 6.2% for silicate and 5.1% for phosphate (Tanhua et al., 2010).



To investigate oxygen and nutrient trends we use the hydrodata set and recent ship sections to construct time series of annual mean oxygen and nutrient profiles. Oxygen and nutrient data are presented in µmol kg$^{-1}$, with data obtained in different units converted to µmol kg$^{-1}$. Linear trends and their 95%

confidence interval were computed using annual averages (all measurements from one year were attributed to that year, as in earlier investigations; e.g. Czeschel et al., 2015) of the profiles linearly interpolated to standard vertical depth levels. As historical measurements are focussed on the upper ocean and as oxygen and nutrient changes will have the largest impact on the biology in the upper ocean, the trends have been computed for the subsurface layer 50 to 300 m as presented in Czeschel et al. (2015). The upper boundary at 50 m was selected to avoid influence from atmosphere-ocean

interaction in the mixed layer. In the North Pacific seasonal variability will reach below 50 m depth. The two areas we consider in the North Pacific contain a lot of data and a test omitting the winter months measurements (January to March: Supplementary Table S1) shows similar trends as for the entire year (Table 1), hence the seasonal cycle did not have a larger impact on the results. The oxygen and nutrient distribution at 50 to 300 m depth (Fig. 2) shows the large variation of the parameters across

the Pacific. The subtropical gyres are clearly visible by enhanced oxygen and low nutrient content. Higher nutrients are seen in the equatorial and eastern Pacific in the areas of equatorial and coastal upwelling, respectively.  As gradients are low in regions of low nutrient content and nutrient data are often sparse, it makes sense to investigate changes in regions with enhanced nutrient content and a sufficiently long time series.

The data used for the oxygen and nutrient time series were interpolated with an objective mapping scheme (Bretherton  et al. 1976) with Gaussian weighting using a temporal half folding range of 0.5 year and a spatial one of 50 m, a maximum temporal range of 1 year and spatial range of 100 m.  The covariance matrix was computed from 100 local data points and 50 random data points within the maximum range, for the diagonal of the covariance matrix a signal to noise ratio of 0.7 was set (see

Schmidtko et al 2013, for details). Due to the random data points the computed trends and confidence intervals vary slightly in each computation run of the interpolation, however the variation is small compared to the confidence interval.





The correlation coefficient between the oxygen time series and the PDO or NPGO were computed for the years after 1976 using the MatLab routine corrcoef and the correlation was considered as significant for a P-value ~<0.05. For some regions with a longer time series the correlation was also computed for temperature or nitrate. In the North Pacific the correlation of oxygen with a 18.6 year oscillating signal

were computed for the entire period with existing oxygen data after removing the derived long-term trends. To investigate the lag the measurement years were shifted from -15 to +15 by 1-year steps. Since the PDO or NPGO time series are continuous, data point reduction due to lag shift are small and only occurring for individual data collected in the last 15years. The impact is assumed to be smaller than the here given uncertainties.

## 3 Trends and influence of climate signals

### 3.1     Long-term trends

In the global ocean the long-term surface temperature trend 1901-2012 was positive everywhere except

for a region in the North Atlantic (IPCC 2013, Fig. 2.21). However, for 1951 to 1980 decreasing surface temperatures were reported across the North Pacific Ocean (IPCC 2013, Fig. 2.22). For 1981 to 2012, while the western Pacific showed a warming trend, a large region with decreasing surface temperatures was seen in the eastern Pacific Ocean. However, a different pattern emerges when the analysis is applied to the subsurface layer (50 to 300 m). Subsurface temperature trends computed for all data since

1950 in each of the regions discussed in section 2, showed weak and not significant temperature increases. Exceptions are 1) the area P with a significant temperature increase of $0.0083 \pm 0.0073°C\ yr^{-1}$ for the period 1954 to 2017, with the highest temperatures in the positive PDO periods 1977 to 1999 and since 2013, and 2) the Oyashio region, where a strong and significant temperature decrease of -$0.0273 \pm 0.0188°C\ yr^{-1}$ was derived for the period 1952 to 2017 with the lowest temperatures in the

period 1977 to 2010 (Suppl. Fig. S2). This agrees with the surface layer (0 to 50 m), where all areas showed increasing temperature for the entire measurement period, except for the Oyashio region where temperature decreased (Suppl. Fig. S2). The very strong temperature increase at area P after 2013 was



impacted by the strong surface temperature anomaly in the Northeast Pacific Ocean during 2013 to 2015 marine heatwave nicknamed The Blob (Bond et al., 2015: Di Lorenzo and Mantua, 2016). For the subsurface layer discussed here, however, it appears that the marine heatwave peaked in 2016-2017 (Jackson et al., 2018). Except for the Aloha and the CALCOFIc regions all 0 to 50 m trends were not

significant, probably due to interdecadal, seasonal and regional variations in the temperature measurements.

In the open ocean a decline in mean oxygen solubility of ~5 μM associated with a hypothetical warming of 1°C throughout the upper ocean would expand the reach of hypoxic conditions by 10% while suboxic zones would nearly triple in volume (Deutsch et al., 2011). However the sensitivity of hypoxic zones to

variations in the depth of the thermocline introduces a mechanism for counteracting this expansion (Deutsch et al., 2011). In the areas of the equatorial Pacific (Fig. 3 and 4), the CalCOFIc region off California (Fig. 5), the area P in the North Pacific (Suppl. Fig. S3), the Aloha region north of Hawaii (Suppl. Fig. S4), the Oyashio region in the western North Pacific (Fig. 7) and a region off Peru (Suppl. Fig. S5), the linear trend of the oxygen content of the layer 50 to 300 m decreases for the entire time

period of available data since 1950, except for the eastern equatorial region 2° to 5°S, 84° to 87°W (just northwest of the Peru region) which has a positive trend caused by low and variable oxygen content in the 1955 to 1965 period. However, except for the western equatorial areas and area P, the oxygen trends are not significant with regard to the entire time series (Table 1).

The long-term trends of the western equatorial regions between 5°N and 5°S at 165°-175°W and 105°-

115°W show a continuous oxygen decrease since 1950 for the 50 to 300 m layer of $-0.36 \pm 0.22$ μmol kg$^{-1}$ yr$^{-1}$  and $-0.65 \pm 0.37$ μmol kg$^{-1}$ yr$^{-1}$  (Fig. 3; Table 1). These trends are larger than the trends for the same regions since 1960 for the 300 to 700 m layer of $-0.19 \pm 0.20$ μmol kg$^{-1}$ yr$^{-1}$ and $-0.13 \pm 0.32$ μmol kg$^{-1}$ yr$^{-1}$ (Stramma et al., 2008), as the deeper layer is located at the low oxygen core of the OMZ where no large changes are possible.

For the period 1956 to 2006 at area P in the depth range 100 to 400 m an ocean warming of 0.005-0.012°C yr$^{-1}$ has been described (Whitney et al. 2007). Our longer temperature time series 1954 to 2017 at 50 to 300 m of 0.0083°C yr$^{-1}$ confirms this trend (Suppl. Fig. S2). The oxygen trend of $-0.24 \pm 0.23$



µmol kg$^{-1}$ yr$^{-1}$ in area P (48°-52°N, 143°-147°W) for the 50 to 300 m layer for 1954 to 2017 is smaller than that previously reported for area P at different depth layers between 100 and 400 m for the shorter time period 1956 to 2006 (0.39-0.70 µmol kg$^{-1}$ yr$^{-1}$; Whitney et al., 2007). For yet another time period (1987 to 2011) and the layer 100-500 m in area P, trends have been reported (Whitney et al., 2013): -0.9

µmol l$^{-1}$ yr$^{-1}$ for oxygen (= density*µmol kg$^{-1}$ yr$^{-1}$; ~0.88 µmol kg$^{-1}$ yr$^{-1}$), 0.085 µmol l$^{-1}$ yr$^{-1}$ for nitrate, 0.30 µmol l$^{-1}$ yr$^{-1}$ for silicate and 0.0033 µmol l$^{-1}$ yr$^{-1}$ for phosphate). Our trends for 50 to 300 m for the longer time period 1977 to 2017 (Table 1) are much smaller for oxygen (-0.18 µmol kg$^{-1}$ yr$^{-1}$) and slightly smaller for the nitrate (0.093 µmol kg$^{-1}$ yr$^{-1}$), silicate (0.193 µmol kg$^{-1}$ yr$^{-1}$) and phosphate (0.001 µmol kg$^{-1}$ yr$^{-1}$).

The Aloha region is located in the southern part of the North Pacific subtropical gyre where oxygen is high and the nutrient inventory low. The oxygen trend since 1951 is negative although not significant. Nitrate and silicate trends (available since 1984/1985) are positive, while phosphate decreases. However, only the trends in silicate (including one measurement in 1970) and phosphate are significant (Table 1).

For a region similar to our Oyashio region an oxygen decrease of 0.73 µmol kg$^{-1}$ yr$^{-1}$ between 1968 to 1998 has been reported for the density layer 26.8-27.4 kg m$^{-3}$ (~260-1030 m), superimposed with a bi-decadal trend of about 18 years (Watanabe et al., 2003). Our oxygen trend for the layer 50 to 300 m is slightly positive (+0.15 µmol kg$^{-1}$ yr$^{-1}$) for the period 1977-2017 and negative (-0.23 µmol kg$^{-1}$ yr$^{-1}$) for the longer time period of 1952 to 2017. The phosphate increase in the deep layer 26.8-27.4 kg m$^{-3}$ was

reported as 0.004 µmol kg$^{-1}$ yr$^{-1}$ in Watanabe et al. (2003) while in our 50 to 300 m layer the phosphate trends for the positive and negative PDO periods and the entire period are larger, up to 0.010 µmol kg$^{-1}$ yr$^{-1}$ (Table 1). We observe a large silicate increase in the Oyashio region, +0.667 µmol kg$^{-1}$ yr$^{-1}$ since 1981 (Table 1). This exceptional silicate enrichment in the Oyashio region was noticed before and it was speculated that warming and a reduction of dense water formation causes this enrichment, because

less silicate is transported into deep (>300 m) waters (Whitney et al., 2013).



In the Peru region the oxygen, silicate and phosphate in the 50 to 300 m layer decrease while nitrate increases (Suppl. Fig. S5). However, these long-term trends are not significant (Table 1), possibly due to the paucity of data and the reversal of trends related to the PDO phases as described below.

The oxygen time series from the different Pacific regions generally show decreasing oxygen, although with varying magnitudes. These variations could be related to different climate signals as investigated below. The nutrient time series often show an increase over time: as expected, this is the opposite trend to oxygen.

## 3.2 The influence of the Pacific Decadal Oscillation (PDO)

An influence of the PDO has been seen in oxygen measurements (e.g. Czeschel et al., 2012) and modelling studies (e.g. Frölicher et al., 2009). According to the description of the PDO influence on the thermocline depth (e.g. Deutsch et al., 2011; Chavez et al., 2003) it is expected that during cold PDO phases the oxygen will decrease and the nutrients increase in the eastern equatorial and tropical Pacific, while during warm PDO periods the oxygen should increase and the nutrients decrease. However, visual inspection of the oxygen time series in the equatorial Pacific (areas E and D; see Fig. 2 in Stramma et al., 2008) indicates stagnant oxygen concentrations before 1976 during the cold PDO and enhanced oxygen depletion since the 1980's in the OMZ in the subsurface layer. The annual mean oxygen concentration for the layer 50 to 300 m for 5°S-5°N, 165-175°W (area E) and 5°S-5°N, 105-115°W (area D) show a strong oxygen decrease after 1976 (Fig. 3a and Fig. 3b). As nutrient data in the open Pacific are sparse, no nutrient trends could be derived for areas E and D. The correlation of the PDO with the 50 to 300 m oxygen is high and significant for all areas along the equator with the highest correlation coefficient of +0.82 at 5°N to 5°S, 165°W to 175°W (area E) and slightly decreasing towards the east (Table 2).

Czeschel et al. (2015) showed decreasing oxygen and increasing nutrients in the 50 to 300 m layer of the area 2-5°S, 84-87°W (area G) since 1976. If PDO influence acts as expected, an even stronger gradient should exist prior to 1976. For the layer 50 to 300 m between 2°S and 5°S, 84°W and 87°W, for the post-1976 positive PDO period, the computed trend is slightly modified compared to the time period investigated in Czeschel et al. (2015). Here we see decreasing oxygen concentrations but



increasing nitrate and phosphate and decreasing silicate concentrations since 1977 (Fig. 4). The main differences between these time series are improvements to the objective analysis, less smoothing being applied, and two more cruises added for 2015 and 2017. The resulting trends since 1977 for oxygen and nitrate are smaller, for phosphate larger (though not significant) and for silicate reversed compared to

Czeschel et al. (2015). Oxygen for the negative PDO phase (1955 to 1976) showed a significant positive trend of $1.63 \pm 1.18$ µmol kg$^{-1}$ yr$^{-1}$ for the 50 to 300 m depth layer in the area G.

Near California the CalCOFIc bottle data in the region 34-35°N, 121-122°W (Fig. 5, Table 1) show similar trends as for the equatorial area (2-5°S, 84-87°W; area G) for the period since 1977, with decreasing oxygen and increasing nitrate, phosphate and silicate. The long-term trend over all

measurements since 1950 show increasing nitrate and phosphate and decreasing oxygen and silicate for the CalCOFIc region similar to the long-term trends for the equatorial region. The correlation between the 50-300 m annual means and the PDO annual mean is 0.44 for oxygen (Fig. 6a; Table 2) and -0.47 for nitrate (Table 2) for the period after 1976 with the PDO lagging by one year.

For the area P (48°-52°N, 143°-147°W) the correlation with PDO (0.33; Table 2) is not significant for

oxygen, while the correlation coefficients of -0.38 for nitrate and of +0.38 for temperature (with the PDO lagging by 2 years) are significant. Also remarkable is that the trends in oxygen are unchanged, whether fitting to the entire record or to the positive and negative PDO time periods separately. Hence in this northern Pacific region the PDO has a weak influence on the 50 to 300 m biogeochemistry, probably caused by water masses propagating by 5 to 15 years from the Oyashio region into this part of

the North Pacific (Ueno and Yasuda, 2003).

The Aloha region is located at the transition of the warm and cold area of the PDO, thus one might expect the influence of the PDO to be weak in this region. However, the oxygen was observed to increase during cold PDO and the trend was close to zero in the warm PDO phase (Suppl. Fig. S4). During the PDO warm phase nitrate and silicate increased while phosphate decreased. There are too few

nutrient data in the period of the PDO cold phase and only phosphate is available with a minor increase (Table 1).



In the global PDO distribution (Suppl. Fig. S1, top) the Oyashio region in the western North Pacific is located in a reversed temperature anomaly pattern compared to the eastern Pacific, so one might expect to see the opposite trends. The oxygen decreases during the cold PDO phase and slightly increases during the warm PDO phase. The nutrients increased during both PDO phases except for silicate which

decreased during the cold PDO phase (Fig. 7, Table 1). Different to the other regions, the temperature of the 50 to 300 m layer shows a significant decrease in this region (Suppl. Fig. S2). The oxygen and nitrate of the Oyashio region show significant correlations with the PDO (Table 2) with the PDO lagging by 4 to 5 years, indicating a delay between the surface signal and the changes in the subsurface layer. In a previous study where oxygen concentrations were investigated from 1954 to 2014, a decrease

of oxygen was attributed to a reduction of ventilation in winter due to warming and freshening and reduction of dense water formation in the Sea of Okhotsk (Sasano et al., 2018).

In the Peru region there is an increase in oxygen in the 50 to 300 m layer during the cold PDO phase and a decrease during the warm PDO phase. Nitrate, silicate and phosphate show trends opposite to oxygen. Despite the paucity of data in this region, these PDO related trends are within the 95%

confidence limit for oxygen in the cold PDO phase, and for nitrate and phosphate since 1977 (Table 1). The correlation of the PDO with the 50 to 350 m oxygen is high (r=+0.64). This indicates that the observed changes in oxygen and nutrients off Peru are associated with the PDO.

Although the time period since the shift from negative to positive PDO in 2013 is short, almost all areas examined here show higher 50 to 300 m oxygen concentrations than the trend line for the period since

1978 and lower nitrate and silicate concentrations then the trend line since 1977, except for the Aloha region, the oxygen in 2017 in the Peru region, and the El Niño year 2015/16 and The Blob at area P.

Figure 1 shows a global mean temperature increase before 1945, a stagnant temperature trend during the PDO cool phase between 1945 and 1976 and a temperature increase after 1976 despite this period encompassing a PDO warm and cold phase. As the influence of the warm PDO phase 1977 to 1999 and

the cold phase 1999 to 2014 is not related to major oxygen and nutrient trend changes, the increasing temperature seems to be a major component of setting the long-term oxygen trend in the Pacific Ocean.



### 3.3    The influence of the North Pacific Gyre Oscillation (NPGO)

As the NPGO is defined for a smaller region in the northeast Pacific than the PDO, its largest influence on oxygen and nutrients is expected to be in the North Pacific. The NPGO index shows higher variability than the PDO index (e. g. Fig. 4). Strong NPGO minima were present in the years 1967,

1980, 1994, 2006 and 2015 and maxima in the years 1961, 1976, 1988, 2000 and 2010.

In the area P, the correlation with NPGO is -0.35 for oxygen in the 50 to 300 m layer for the period since 1977 with NPGO leading by 4 years; and +0.61 (and significant) for nitrate with NPGO lagging by one year (Table 2). The oxygen data are highly variable which might have led to the low correlation with the NPGO, however the nitrate correlation as well as a correlation of NPGO with temperature of -

0.65 show a strong relationship with the NPGO in this region.

Since 1980, the maxima and minima of the CalCOFIc time series of nitrate, phosphate and silicate (Fig. 5) often agree with the NPGO maxima and minima. The correlation between the 50-300 m annual mean and the NPGO annual mean is -0.33 for oxygen (Fig. 6b) and +0.38 for nitrate with the NPGO lagging by 1 year. The correlation with the NPGO is stronger than with the PDO in the North Pacific at area P

and weaker at CalCOFIc. This confirms the described decadal variations linked to the NPGO in the North Pacific and the California Current System (Di Lorenzo et al., 2009).

In the Oyashio region, oxygen shows a significant correlation of +0.44 with the NPGO, while the correlation of nitrate and temperature with the NPGO are weaker and not significant. At the southern part of the North Pacific subtropical gyre, in the Aloha area, the NPGO correlation coefficient of +0.56

with oxygen is high and larger than the PDO correlation. In the central equatorial Pacific and off Peru the correlation of oxygen with NPGO is significant but lower than the correlation with the PDO.

### 3.4    The influence of the Sub-Tropical Cells (STC), the North Pacific Index (NPI) and El Niño-Southern Oscillation (ENSO)

The strongest influence of the STC is expected in the tropical Pacific. For negative PDO phases

tropical-extratropical interactions should lead to reduced mass flux in the STC and a reduction in the eastward flowing Equatorial Undercurrent (Henley, 2017). The reduced STC water transport from the



1960s to the 1990s should lead to reduced oxygen and nutrient transports to the tropics. After this, increased STC should lead to increased oxygen and nutrient transports in the early 2000s. Duteil et al. (2014) state that the simulated decrease in strength of the STC of about 30% during the 1960s to 1990s is in good agreement with observations and should induce a decline in tropical ocean oxygen and

phosphate concentrations and a pause in oxygen decrease in the near future. The oxygen concentration at 50 to 300 m decreases at the equator at 165° to 175°W (area E) and 105° to 115°W (area D) over the entire record (1950-2017), while at 2° to 5°S, 84° to 87°W (area G) oxygen increases until the late 1970s and then decreases until 2017 (Fig. 3). However, at 165° to 175°W (area E), and 84° to 87°W (area G) the oxygen concentration increases after 2000 and this might be a signal related to the STC.

The changes of the trend in oxygen and nutrients in the late 1970s in the eastern equatorial region (Fig. 3 and 4) suggest that these trends are not well correlated to STC changes. Due to the long duration of the STC phases and the sparse data set, it is not possible to perform a meaningful correlation analysis to investigate STC influence on the oxygen and nutrient variations.

The bi-decadal oscillation related to NPI with minima in 1962 and 1983 and maxima in 1971 and 1991

(Watanabe et al., 2003, their Fig. 2) is difficult to see in our analysis of the Oyashio region due to large year-to-year variability (Fig. 7). The correlation of the NPI (November to March anomaly) with the 50 to 300 m oxygen time series since 1977 leads to significant correlations of -0.30 at the Oyashio region and 0.44 at area P, without a time lag at either locations.

A bidecadal oscillation of 16.4-19.6 years in oxygen, possibly driven by nodal tidal cycles of 18.6 years,

was described recently for the Oyashio region with maxima at about 1971, 1989 and 2008 and minima at about 1962, 1980 and 1998 (Fig. 3 in Sasano et al., 2018). The 50 to 300 m oxygen time series does not show a strong visual correlation for the period 1961 to 2008 between the trend-corrected oxygen and an 18.6 year oscillation. At area P oxygen trends include periods of increased ventilation of deeper isopycnals on a ~18 year cycle (Whitney et al., 2007). The correlation of the detrended oxygen content

of the 50 to 300 m layer for the period 1954 to 2017 with a 18.6 year cycle was significant with r = -0.27. The correlation with the 18.6 year oscillation is similar (0.25) at area P, but is weak and not significant at both the CalCOFIc and the Aloha areas.



Similar parameter distributions as for cold PDO periods exist for La Niña events and as for warm PDO periods for El Niño events (Deutsch et al., 2011). Surprisingly in the equatorial regions (Fig. 3 and 4) the subsurface oxygen concentration at 50 to 300 m depth shows no clear anomalies in years of ENSO events. The ENSO signal seems to be restricted to the near surface layer in the equatorial Pacific. In the

eastern Pacific during El Niño periods, oxygen in the upper ocean is higher and nutrients are lower in the upwelling regions (CalCOFIc and Peru region) due to either reduced upwelling or upwelling of oxygen-richer and nutrient poorer water masses. In the CalCOFIc region, the measurements during the very strong El Niño events in 1997/98 and 2015/16 show higher oxygen concentration and very low nutrient concentration in the 50 to 300 m layer when compared to the trend-line and the neighbouring

years (Fig. 5). The deviations are very strong for the 1997/98 El Niño while moderate for the 2015/2016 El Niño. This signal is also visible for the 2015/2016 El Niño in the Peru region (Suppl. Fig. S5) and for the 1997/98 El Niño at a shelf station off Lima (Graco et al., 2017). Not all strong El Niño events are associated with similar anomalies. Offshore from the upwelling region, at area P, the anomalies for the very strong El Niños 1997/98 and 2015/2016 are opposite, with low oxygen and high nutrient

concentrations during the earlier event (Suppl. Fig. S3). For La Niña events a reversed trend is visible in the eastern Pacific for some events, e.g. with low oxygen and high nutrient concentrations for the 1988/1989, the 1998/1999 and the 1999/2000 La Niña events in the CalCOFIc region (Fig. 5). These ENSO related signals disappear in the following year and hence the ENSO related changes in oxygen and nutrients do not show a multi-year signal.

**4 Discussion**

One might wonder if the observed changes in trends in oxygen and nutrients are more strongly influenced by evolving methods rather than changes in climate. While the Winkler titration method to measure oxygen has remained the same since the early 1900's, the methods to determine nutrients have varied. However, except for the very low nutrient data in the Aloha region in the center of the North

Pacific, all areas used here show a similar and relatively large range for nitrate, silicate and phosphate. Precision of 5% for nitrate measurements of more than 10 µmol L$^{-1}$, of about 6% or less for silicate and 5% for phosphate at 0.9 µmol L$^{-1}$ and larger are reported for early nutrient measurements (Hansen and



Koroleff, 1999). Similar offsets for measurements after the 1990's were derived with 3.5% for nitrate, 6.2% for silicate and 5.1% for phosphate (Tanhua et al., 2010), accordingly the offsets for the eastern equatorial region (Fig. 4) could be as high as ~1 µmol kg$^{-1}$ for nitrate, ~0.1 µmol kg$^{-1}$ for phosphate and ~1.0 µmol kg$^{-1}$ for silicate and hence smaller than the observed long-term trends.

To put the regions examined here in context, we compare with previously published trends for the entire North Pacific as well as changes influencing the ocean circulation. Yasunaka et al. (2016) reported trends of surface phosphate and silicate averaged over the North Pacific from 1961 to 2012 as -0.012 ± 0.005 µmol l$^{-1}$ decade$^{-1}$ and -0.38 ± 0.13 µmol l$^{-1}$ decade$^{-1}$, respectively, whereas the nitrate trend averaged over the North Pacific was 0.01 ± 0.13 µmol l$^{-1}$ decade$^{-1}$. This is in contrast to the subsurface

layer examined here, where nitrate tended to increase over a similar time period. In particular, high nitrate increase was observed for the Oyashio region with +0.143 µmol kg$^{-1}$ yr$^{-1}$ for the period 1977 to 2017 (Table 1). An increase in anthropogenic nitrogen emissions from northeastern Asia and subsequent deposition over the North Pacific resulted in a detectable increase of nitrate concentrations in the near surface layer since the 1970s (Kim et al., 2014). The observed nitrate increase at 50 to 300 m

might be the response of decreased water subduction. For example, the North Pacific Intermediate Water is a dominant pathway to enter the mid depth waters in the North Pacific and was freshening in the period 1960 to 1990 (Wong et al., 2001). Since the overturning in the North Pacific originates from the Okhotsk Sea through dense shelf water the observed freshening to depth of ~500 m during the past four decades could possibly weaken the shallow overturning of the North Pacific (Ohshima et al.,

2014). Thus interior ocean circulation is likely slowing because less dense water is forming in the Sea of Okhotsk (Sasano et al., 2015, 2018) and hence decreasing the supply of oxygen.

Model results indicate that more than 50% of the total internal variability of oxygen is linked to the PDO in the North Pacific surface and subsurface waters (Frölicher et al., 2009). The long-term 50-300 m trends since the 1950s in the eastern equatorial Pacific and the CalCOFIc region (Fig. 4 and 5; Table

1) indicates a long-term increase in nitrate and phosphate and a decrease in silicate, but often with reversed trends in oxygen and nutrients when separated into cold and warm PDO phases. From the 1980s to the 2010s in the North Pacific, oxygen decreased while nitrate, phosphate and silicate increased (Whitney et al., 2013), similar to what was observed in the eastern equatorial Pacific and the

CalCOFIc region for the period after 1976 (Fig. 4 and 5). For the California Current system the decadal oxygen changes seem to be primarily controlled by ocean circulation dynamics (Pozo Buil and Di Lorenzo, 2017). Pozo Buil and Di Lorenzo (2017) found that subsurface anomalies in the core of the North Pacific Current propagate the oxygen signal downstream within about 10 years to the coastal

regions, and predicted a strong decline in oxygen by 2020 in the California Current system. The recent measurements in the CalCOFIc region, shown here, support this prediction.

The nutrient increase in the CalCOFIc region since the 1980s could be related to upwelling variability in the California Current. A strong nitrate flux from 1980 to 2010 was driven almost entirely by enhanced equatorward winds, negating a weak negative trend associated with increased surface heat

flux (Jacox et al., 2015). However, changes in the properties of source waters (primarily from the eastern tropical Pacific via the California Undercurrent) have likely driven most of the biogeochemical trends observed in the southern California Current (Meinvielle and Johnson, 2013; Bograd et al., 2015; Nam et al., 2015).

Despite the low data coverage, the measurements in the Peru region confirm the expected opposite

trends for oxygen and nutrients related to the PDO-phases. For a shallow shelf station (145 m depth) near Lima, measurements in the upper 100 m show increasing oxygen concentrations for the period 1999 to 2011 (Graco et al., 2017). This contrasts with the decreasing oxygen trend we observe from 1977 to 2017 in the Peru region (Suppl. Fig. S5, Table 1) and indicates that different processes and trends might exist on the shelf compared to the open ocean.

In the equatorial Pacific time series no clear signal of the STC on oxygen and nutrients was observed. In the eastern tropical and subtropical Pacific very strong El Niño and some strong La Niña events are apparent in the oxygen and nutrient distribution but do not result in a multi-year signal or trend.

## 5 Concluding remarks

In this study, we investigated the influence of well-documented atmosphere-ocean decadal oscillations

on the trends in oxygen and nutrients in the upper subsurface layer of the Pacific Ocean. Due to the limited subsurface nutrient data, only select areas could be investigated. Especially in the South Pacific



data are sparse, in part because not all existing data have been made public. A test excluding the winter months (January to March) in the area P and the Oyashio region (Supplementary Table S1) showed that the seasonal cycle had little influence on the trends derived for the 50 to 300 m layer. The depth layer of 50 to 300 m was selected as this is the major layer of biological subsurface activity influenced by

oxygen and nutrient variability. For example, several warm-water mesopelagic species in the California Current, which are apparently adapted to the shallower, more intense OMZ off Baja California, were shown to be increasing despite declining midwater oxygen concentrations and becoming increasingly dominant (Koslow et al., 2018). Enhanced biological activity in coastal regions might lead to larger nutrient variability and obscure climate related signals.

Agreeing well with the regions with the largest SST signal of the PDO (Suppl. Fig. S1a), the PDO seems to have the strongest influence in the 50 to 300 m layer in the equatorial and eastern Pacific (Table 2). During the cold PDO phase in the eastern Pacific and the stagnant global surface temperature signal, the CalCOFIc region and the Peru region the 50 to 300 m oxygen increases and the nutrient concentrations decrease (Fig. 5, Suppl. Fig. S5) and show the opposite trends during the global warming

period since 1977, which we call the warm PDO phase despite a period of a PDO cool phase (Figure 1). An increase of oxygen from 1950 to 1980 and a decrease after 1980 has also been described for some isopycnal surfaces in subsurface waters of the Northeast Pacific (Crawford and Peña, 2016).

With respect to other climate indices, the results are more mixed. The NPGO has the largest impact on decadal variations in the North Pacific at area P (Table 2). The NPGO influence is also visible in the

central and eastern North Pacific and the equatorial Pacific, although weaker than the correlation with the PDO. The NPI is correlated to the oxygen changes in the Oyashio region and area P, without a time lag at either location. The 18.6 year nodal tidal cycle has the largest correlation in the Oyashio region and the area P but is not visible in the CalCOFIc and the Aloha regions. In the Oyashio region a combination of the PDO, the NPGO, the NPI and the 18.6 year nodal cycle contributes to the trends in

temperature, oxygen and nitrate. The oxygen decline in the Oyashio region has different controlling mechanisms for different depth layers. In the upper layer above $\sigma_\theta$=26.7 kg m$^{-3}$ (~200 m) the oxygen decline is primarily attributed to the reduction of winter convection upstream and in part to the deepening of isopycnal surfaces due to warming and freshening in the upper layers, while below



$\sigma_\theta$=26.7 kg m$^{-3}$ the oxygen decline is attributed to the reduction of Dense Shelf Water formation in the Sea of Okhotsk associated with the reduction of sea ice production and freshening (Sasano et al., 2018).

The PDO and NPGO strongly influence the trends in oxygen and nutrient inventories; nevertheless the long-term sea surface temperature trend (Fig. 1) seems to play a role in the oxygen trend as long-term
trends indicate an oxygen decrease throughout the cold and warm PDO phases in all but one area. Note that while the most likely basin-wide drivers of oxygen and nutrient variability were investigated here, other contributors might exist depending on the region, as shown in detail for the Oyashio region by Sasano et al. (2018).

As greenhouse gas concentrations rise further, a variable but positive trend of increasing global mean
surface temperature is expected, which should lead to a continuing decrease of oxygen and increase of nutrients in the subsurface layer important for biological activity. These biogeochemical changes can be expected to have significant economic consequences through changes in the availability of living marine resources.

**Data availability**

The NPGO Time series was taken from http:www.o3d.org/npgo/npgo.php on 18 September 2018 with data available for January 1950 to February 2018. The yearly PDO data were taken from http://ds.data.jma.go.jp/tcc/tcc/products/elnino/decadal/annpdo.txt on 18 September 2018 from the Japan Meteorological Society covering the period 1901 to 2017. The NPI November to March anomaly
was downloaded from https://climatedataguide.ucar.edu/climate-data/north-pacific-np-index-trenberth-and-hurrell-monthly-and-winter (status 4 January 2019, covering the period 1900 to 2018). The historical hydrographic data sets used here are as in Schmidtko et al. (2017), the references are listed in their paper in the Extended Data Table 2). The bottle data from cruises in 2016 at 170°W (096U2016426_hyd1.csv) and at 110°W (33RO20161119_hyd1.csv) were downloaded from
https://cchdo.ucsd.edu on 8 November 2018).

The CalCOFI data set downloaded from http://www.calcofi.org/ccdata.html, status 13 August 2018; data period March 1949 to November 2017. Station P data (at 50°N, 145°W) are from the Institute of Ocean Sciences, Sidney BC, Canada (status September 2018) for the time period May 1956 to August 2017.

The Oyashio region data are from hydrodata, and updated data collections used in Whitney et al. (2013) and Sasano et al. (2018). The Aloha station data (at 22°45'N, 158°W were downloaded from http://hahana.soest.hawaii.edu/hot/hot-dogs/bextraction.html status 15 January 2019, time period covered October 1988 to December 2017).

The assembled measurements of the Meteor cruises in February 2009, November 2012, December
2012, March 2017 and June 2017 and the Sonne cruise in October 2015 used in this paper will be made available before final publication.

**A Supplement is related to this article.**

*Author contributions.* L. Stramma conceived the study, wrote the manuscript had been chief scientist on two of the RV *Meteor* cruises and carried out the hydrographic measurements on two other Pacific
cruises. S. Schmidtko handled the large-scale data sets and developed the optimal interpolation. T. Ono and D. Sasano provided the expertise for the Oyashio region, F. Whitney and T. Ross the data set and expertise for the North Pacific and S. Bograd his expertise for the CalCOFI region. All authors discussed and modified the manuscript.

*Acknowledgements.* The Deutsche Forschungsgemeinschaft (DFG) provided support as part of the
"Sonderforschungsbereich 754: Climate-Biogeochemistry Interactions in the Tropical Ocean" and for the RV *Meteor* cruises. We thank M. Robert and Fisheries and Oceans Canada staff for the core measurements on Line P cruises, as well as the innumerable scientists and crew who have contributed to the Line P program since 1956. The Line P ocean monitoring program is funded by Fisheries and Oceans Canada.

We are grateful to Miriam O'Brien who prepared the figure for http://blog.hotwhopper.com/ and her permission to use the figure in this publication.

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




**Table 1.** Linear trends of solutes in µmol kg$^{-1}$ yr$^{-1}$ with 95% confidence intervals where data are available for the entire period since 1950, for negative (1950-1976; PDO-) and positive (after 1976; PDO+) PDO periods in the 50 to 300 m depth layer for selected ocean areas shown in Figures 3, 4, 5, 7, S3, S4 and S5. Trends not within the 95% confidence interval are shown in *italics*. Areas named: CalCOFIc (34°-35°N, 121°-122°W), area P (48°-52°N, 143°-147°W), Aloha region (22°-25°N, 156°-159°W), Oyashio region (39°-42°N, 144°-149°E) and Peru region (7°-12°S, 78°-83°W).

| Parameter | trend   time period | PDO- trend   time period | PDO+ trend   time period |
|---|---|---|---|
| Area | 5°N-5°S, 165°-175°W | 5°N-5°S, 165°-175°W | 5°N-5°S, 165°-175°W |
| Oxygen | -0.36 ± 0.22 1950-2016 | *-0.19 ± 0.78*  1950-1975 | *-0.56 ± 0.57*  1977-2016 |
| Area | 5°N-5°S, 105°-115°W | 5°N-5°S, 105°-115°W | 5°N-5°S, 105°-115°W |
| Oxygen | -0.65 ± 0.37 1957-2016 | *-2.4 ± 3.1*   1957-1971 | -0.84 ±  0.55  1979-2016 |
| Area | 2°-5°S, 84°-87°W | 2°-5°S, 84°-87°W | 2°-5°S, 84°-87°W |
| Oxygen | *+0.17 ± 0.48* 1955-2017 | +1.63 ± 1.18   1955-1976 | *-0.12  ± 0.54*  1979-2017 |
| Nitrate | *+0.068 ± 0.189* 1966-2017 | *-0.903 ± 1.180*   1966-1976 | *+0.065  ± 0.164* 1979-2017 |
| Silicate | -0.079 ± 0.077 1960-2017 | *-0.381 ± 0.545*  1960-1976 | *-0.098 ± 0. 172*  1983-2017 |
| Phosphate | *+0.003 ± 0.010* 1960-2017 | *-0.018 ± 0.044*  1960-1976 | *+0.011 ± 0.026*  1983-2017 |
| Area | CalCOFIc | CalCOFIc | CalCOFIc |
| Oxygen | *-0.18 ± 0.41*  1950-2017 | +1.13 ± 0.96  1950-1976 | -0.58 ± 0.40  1978-2017 |
| Nitrate | *+0.035 ± 0.054* 1969-2017 | *-1.040 ± 4.230* 1969-1973 | +0.066 ± 0.061 1978-2017 |
| Silicate | *-0.021 ± 0.059* 1961-2017 | *-0.215 ± 0.625* 1961-1973 | *+ 0.029 ± 0.091* 1978-2017 |
| Phosphate | *+0.001 ± 0.003* 1950-2017 | *-0.000 ± 0.016* 1950-1973 | +0.006 ± 0.005 1978-2017 |



| Area | area P | area P | area P |
|------|--------|--------|--------|
| Oxygen | -0.24± 0.23 1954-2017 | *-0.16 ± 1.41* 1954-1976 | *-0.18 ± 0.42* 1977-2017 |
| Nitrate | +0.071 ± 0.056 1956-2017 | *-0.113 ± 0.515* 1956-1973 | *+0.093 ± 0.096* 1980-2017 |
| Silicate | +0.492 ± 0.180 1957-2017 | *+1.47 ± 4.92* 1957-1971 | *+0.193 ± 0.261* 1987-2017 |
| Phosphate | *+0.001 ± 0.003* 1954-2017 | *-0.013 ± 0.023* 1954-1971 | *+0.001 ± 0.008* 1980-2017 |

| Area | Aloha region | Aloha region | Aloha region |
|------|--------------|--------------|--------------|
| Oxygen | *-0.08 ± 0.21* 1951-2017 | *+0.20 ± 0.45* 1951-1976 | *+0.004 ± 0.38* 1977-2017 |
| Nitrate | none | *none* | +0.014 ± 0.021 1984-2017 |
| Silicate | +0.013 ± 0.013 1970-2017 | none | *+0.013 ± 0.016* 1985-2017 |
| Phosphate | -0.002 ± 0.001 1953-2017 | *+0.007 ± 0.020* 1953-1966 | *-0.0004 ± 0.002* 1985-2017 |

| Area | Oyashio region | Oyashio region | Oyashio region |
|------|----------------|----------------|----------------|
| Oxygen | *-0.23 ± 0.34* 1952-2017 | *-0.39 ± 0.60* 1952-1976 | *+0.15 ± 0.69* 1977-2017 |
| Nitrate | +0.090 ± 0.068 1964-2017 | *+0.164 ± 0.520* 1964-1976 | +0.143 ± 0.079 1977-2017 |
| Silicate | *+0.176 ± 0.370* 1952-2017 | -1.38 ± 1.09 1952-1972 | +0.667 ± 0.330 1981-2017 |
| Phosphate | +0.006 ± 0.004 1953-2017 | *+0.010 ± 0.015* 1953-1976 | +0.010 ± 0.007 1977-2017 |

| Area | Peru region | Peru region | Peru region |
|------|-------------|-------------|-------------|
| Oxygen | *- 0.05 ±0.32* 1960-2017 | +0.92 ± 0.68 1960-1976 | *-0.34 ± 0.40* 1977-2017 |
| Nitrate | *+0.068 ± 0.216* 1965-2017 | *-1.03 ± 1.35* 1965-1976 | +0.181 ± 0.073 1977-2017 |
| Silicate | *-0.062 ± 0.150* 1965-2017 | *-0.707 ± 1.040* 1965-1976 | *+0.032 ± 0.145* 1977-2017 |
| Phosphate | *-0.000 ± 0.010* 1960-2017 | *-0.006 ± 0.032* 1960-1976 | +0.012 ± 0.010 1977-2017 |



**Table 2.** Correlation coefficient since 1977 between annual layer 50 to 300 m concentration and PDO and NPGO with PDO/NPGO lags (negative PDO/NPGO leads). Correlation not significant (P-value~<0.05) is shown in brackets.

| Area | Parameter | PDO-lag | PDO-correlation | NPGO-lag | NPGO-correlation |
|---|---|---|---|---|---|
| 5°N-5°S, 165°W-175°W | oxygen | -7 | 0.82 | -6 | -0.61 |
| 5°N-5°S, 105°W-115°W | oxygen | -1 | 0.72 | -3 | -0.48 |
| 2°S-5°S, 84°W-87°W | oxygen | 0 | 0.62 | +1 | 0.73 |
| CalCOFIc | oxygen | +1 | 0.44 | +1 | -0.33 |
| CalCOFIc | nitrate | +1 | -0.47 | +1 | 0.38 |
| Area P | oxygen | (-7 | 0.33) | -4 | -0.35 |
| Area P | nitrate | +2 | -0.38 | +1 | 0.61 |
| Area P | temperature | +2 | 0.38 | +1 | -0.65 |
| Aloha region | oxygen | +2 | -0.49 | +2 | 0.56 |
| Oyashio region | oxygen | +5 | -0.44 | +5 | 0.44 |
| Oyashio region | nitrate | +4 | -0.35 | (+5 | 0.29) |
| Oyashio region | temperature | -7 | +0.36 | (-6 | -0.29) |
| Peru region | oxygen | 0 | 0.64 | +1 | -0.41 |



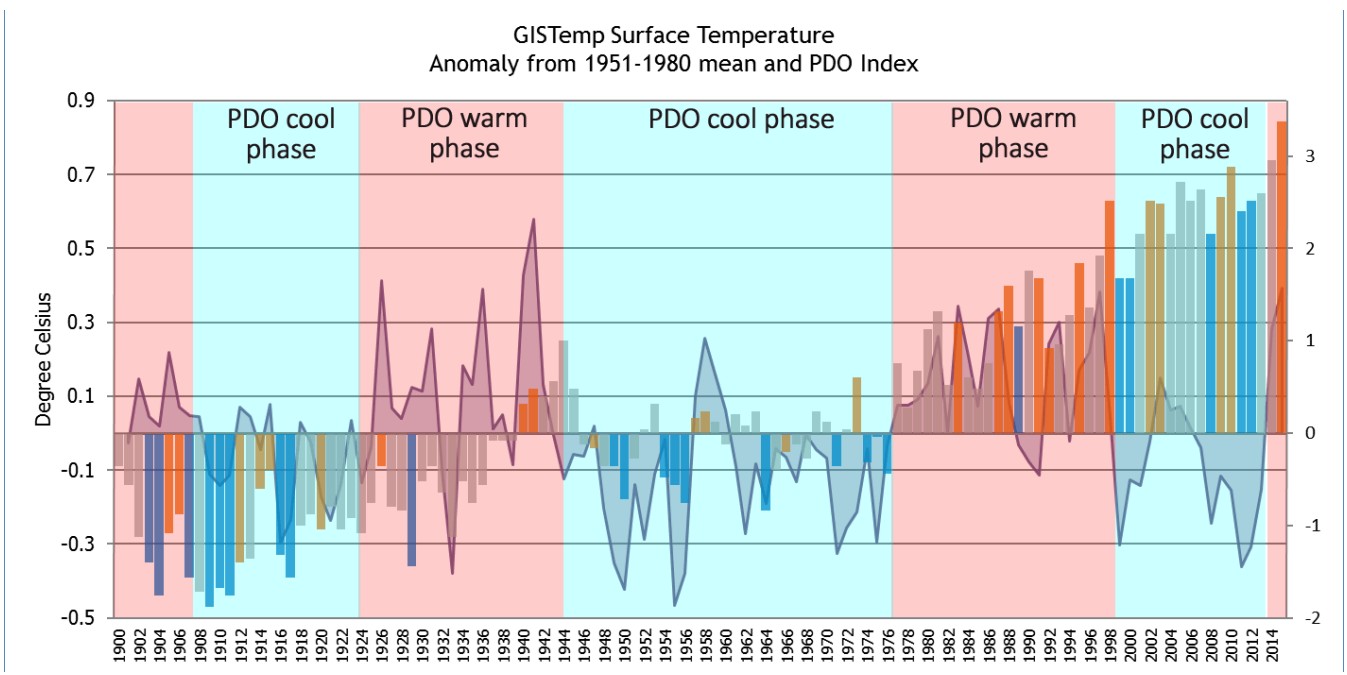

**Figure 1.** Global mean surface temperature anomaly from the 1951 to 1980 mean (GISTemp; peak of the annual bars; left scale) with La Niña years (blue bars), El Niño years (orange/brown bars) neutral years (grey bars) and the PDO index (solid line; right scale), with PDO phases marked (compiled by Miriam O'Brien for 1900 to 2015). The data sources used GISS NASA (temperature), Australian Bureau of Meteorology (ENSO years based on the Southern Oscillation Index) Japan Meteorological Society (PDO index) and Trenberth 2015 (PDO phases).



**Figure 2.** Distribution of the areas used and mean 50 to 300 m silicate, phosphate, nitrate and oxygen (all in µmol kg$^{-1}$), in the Pacific Ocean.





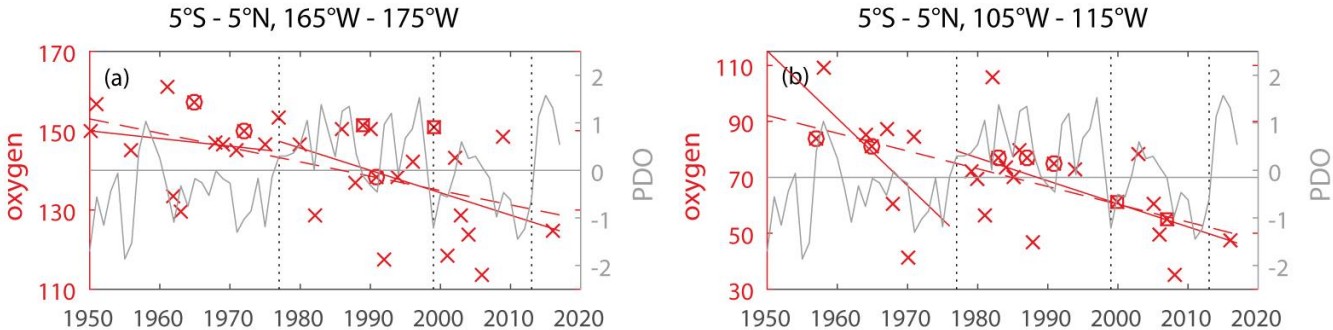

**Figure 3.** Annual mean oxygen concentration (in µmol kg$^{-1}$, red crosses) for years available and trends for the layer 50 to 300 m plotted for the entire time period (dashed red lines) and for the periods 1950 to 1976 for the negative PDO phase and after 1976 for the positive PDO phase (solid red lines) for a) 5°S-5°N, 165-175°W (area E) and b) 5°S-5°N, 105-115°W (area D). El Niño years defined as strong or very strong are marked by an additional circle, strong La Niña years by an additional square. The change of the PDO status in 1977, 1999 and 2013 are marked by vertical dotted lines. The annual mean PDO index is shown as grey curve.





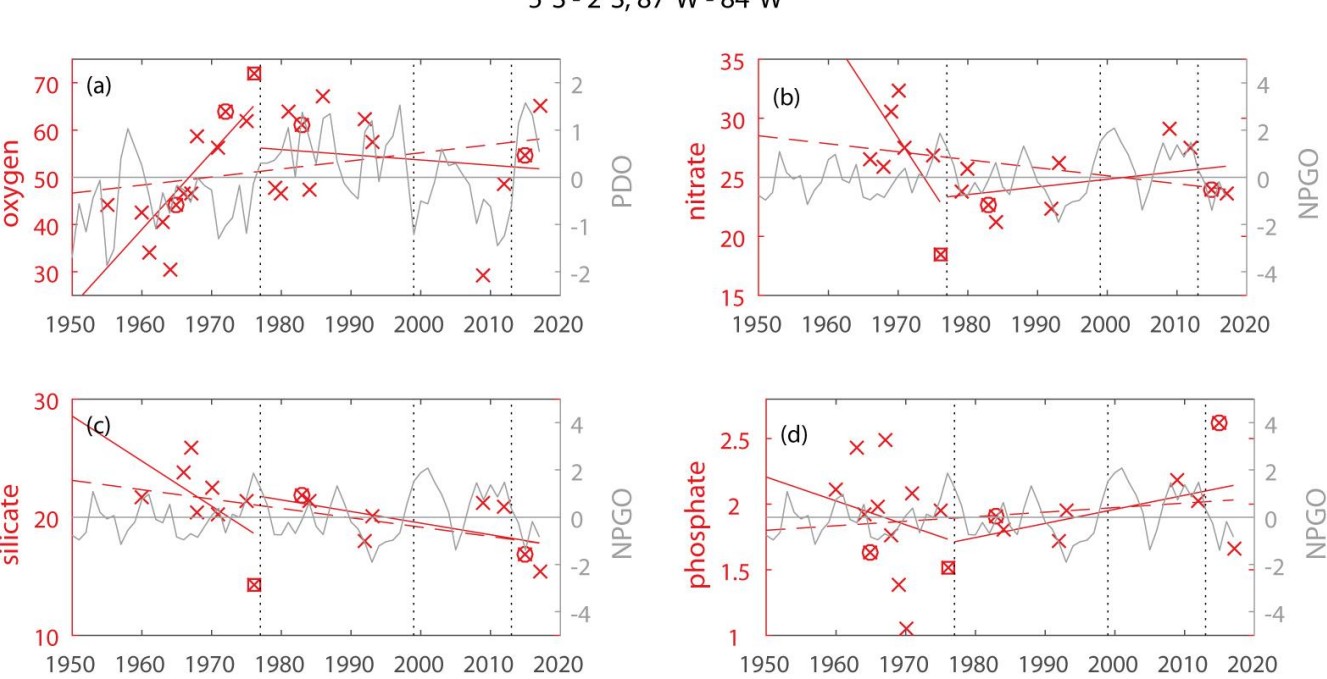

**Figure 4.** Annual mean concentration for years available (see Table 1) and trends for the layer 50 to 300
m plotted for the entire time period (dashed red lines) and for the periods 1950 to 1976 for the negative

5    PDO phase and after 1976 for the positive PDO phase (solid red lines) between 2°S and 5°S, 84°W and
87°W (area G) in µmol kg$^{-1}$ yr$^{-1}$ for oxygen, nitrate, silicate and phosphate. El Niño years defined as
strong or very strong are marked by an additional circle, strong La Niña years by an additional square.
The change of the PDO status in 1977, 1999 and 2013 are marked by vertical dotted lines. The annual
mean PDO index is shown in the oxygen time series as grey curve and the NPGO index is shown in the

10   nitrate, silicate and phosphate time series as grey curve.

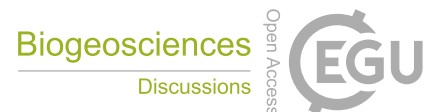

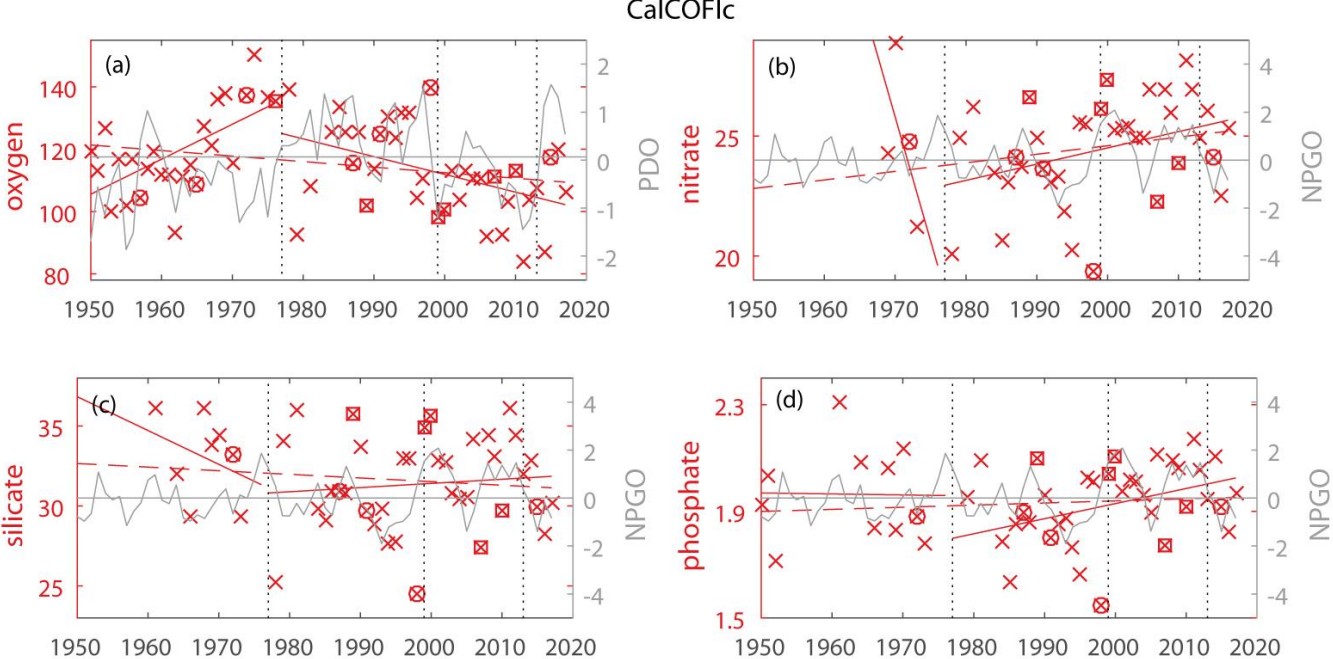

**Figure 5.** Annual mean concentration for years available and trends for the layer 50 to 300 m plotted for the entire time period (dashed red lines) and for the periods 1950 to 1976 for the negative PDO phase and after 1976 for the positive PDO phase (solid red lines) between 34°N and 35°N, 121°W and 122°W from the CalCOFIc bottle data in µmol kg⁻¹ yr⁻¹ for a) oxygen, b) nitrate, c) silicate and d) phosphate. El Niño years defined as strong and very strong are marked by an additional circle, strong La Niña years by an additional square. The change of the PDO status in 1977, 1999 and 2013 are marked by vertical dotted lines. The annual mean PDO index is shown in the oxygen time series as grey curve and the NPGO index is shown in the nitrate, silicate and phosphate time series as grey curve.





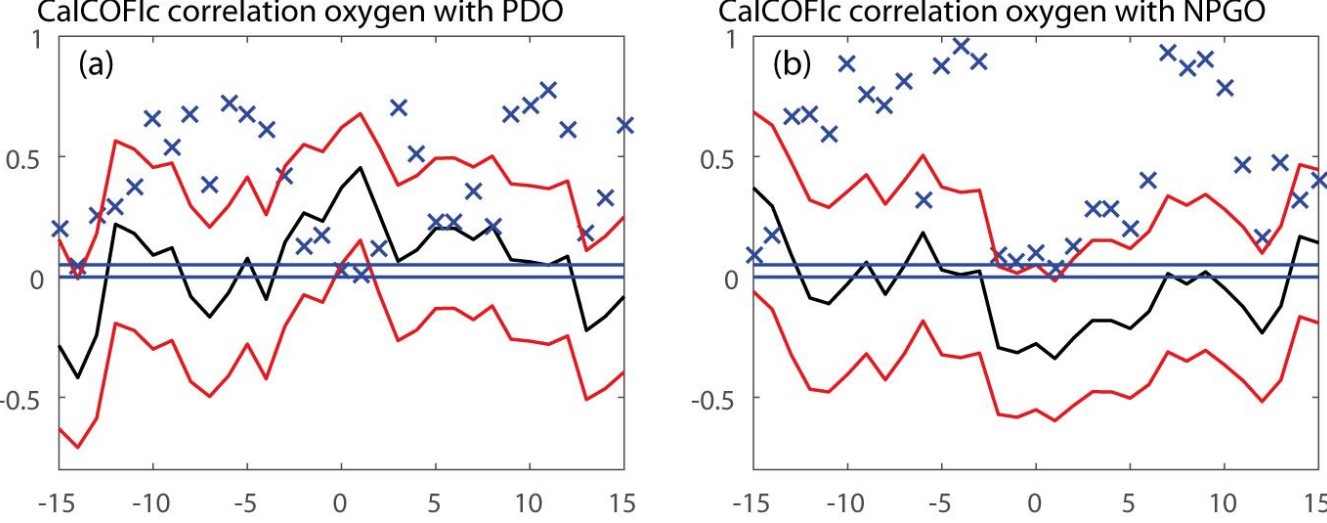

**Figure 6.** Correlation between oxygen at 50-300 m in the region 34-35°N, 121-122°W (CalCOFIc) and
a) the PDO and b) the NPGO (black lines) shifted between -15 and +15 years for the years after 1976
with the lower and upper bounds of the 95% confidence interval (red lines) and the p-values (x) which
shows significant correlation for p ~>0.05 (blue lines for 0 and 0.05). For positive years PDO/NPGO
lags, for negative years they lead.



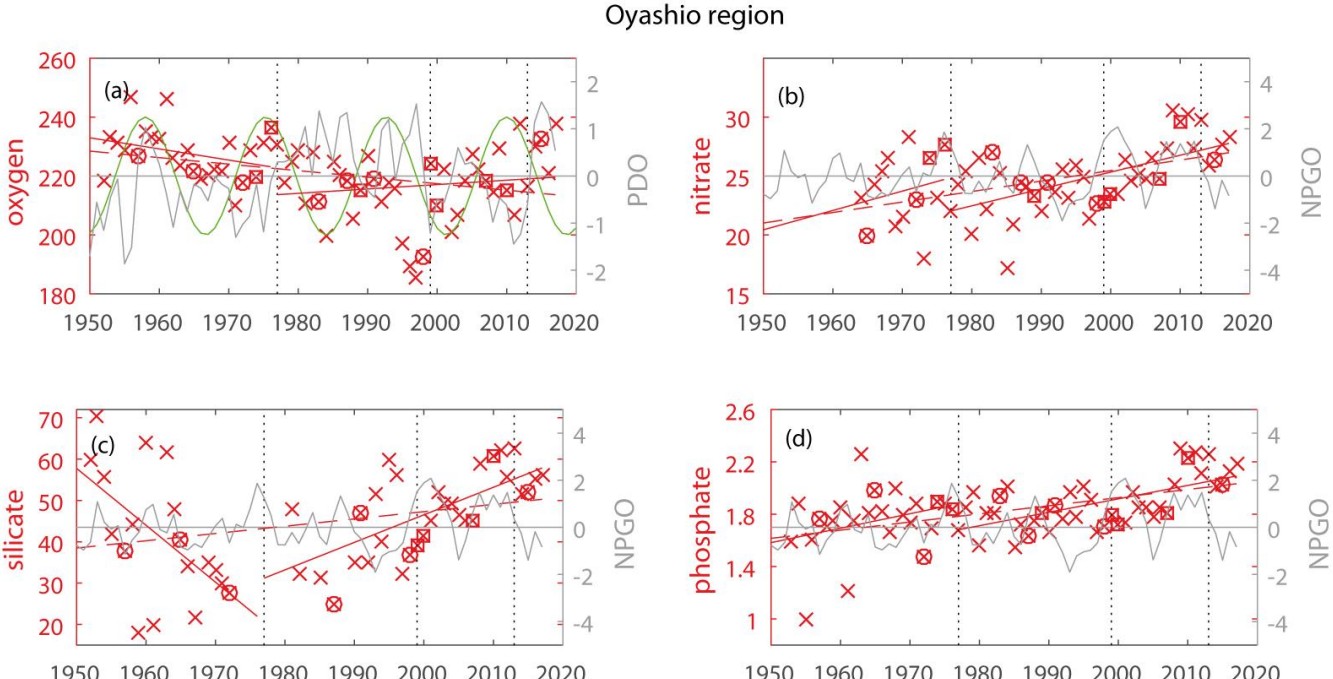

**Figure 7.** Annual mean concentration for years available and trends for the layer 50 to 300 m plotted for the periods 1950 to 1976 for the negative PDO phase and after 1976 for the positive PDO phase (solid red lines) and for the entire time period (dashed red lines) for the Oyashio region (39° - 42°N, 144° - 149°E) from hydrodata CTD and bottle data and a data collection used in Whitney et al. (2013) and Sasano et al. (2018) in µmol kg$^{-1}$ yr$^{-1}$ for a) oxygen, b) nitrate, c) silicate and d) phosphate. For nitrate measurements in 1963 were removed as the 50-300 m mean was much too high (1.27 µmol kg$^{-1}$). El Niño years defined as strong are marked by an additional circle, strong La Niña years by an additional square. The change of the PDO status in 1977, 1999 and 2013 are marked by vertical dotted lines. In the oxygen time series the 18.6 year sinusoidal nodal cycle is included (green curve). The annual mean PDO index is shown in the oxygen time series as grey curve and the annual mean NPGO index is shown in the nitrate, silicate and phosphate time series as grey curve.