# Peer review of "Trends and decadal oscillations of oxygen and nutrients at 50 to 300 m depth in the North and Equatorial Pacific"

_Biogeosciences, 2019_

## Referee Comment (RC1) · Anonymous Referee #1 · 6 May 2019

GENERAL ASSESSMENT

Dr Lothar Stramma and coauthors compiled a mostly public dataset of oxygen, nitrate, silicate and phosphate observations from eight areas of the Pacific Ocean. Using those data, they calculated long-term trends over the entire length of the time series. They also calculated trends over the 1950 to 1976 negative phase of the Pacific Decadal Oscillation (PDO), and after 1976 for the positive phase of the PDO. They showed that in some cases, trend estimates were very different depending on sign of PDO phase. In addition, they calculated lagged correlation coefficients between oxygen, nutrient and temperature time series from their eight areas with the PDO and with the North Pacific Gyre Oscillation (NPGO) climate indices. They also briefly examined how three other modes of climate variability (ENSO, STC, NPI) may or may not affect the time

series in the same eight areas.

This is a worthwhile study. It is important that we better understand "The influence of decadal oscillations on the oxygen and nutrient trends in the Pacific Ocean". But there is a big caveat: the word DECADAL in the paper's title aptly captures the idea that successive years are not statistically independent of each other. For instance, when a given year is above normal for the PDO, the odds of having a below normal value of the PDO the following year is actually lower than 0.5; the dice are rigged. Several consecutive years with above normal values tend to be lumped together, and likewise for below normal values. From the point of view of statistical analysis, this implies the assumption that successive annual values in the oxygen, nutrients and temperature time series are independent of each other is not valid. Consequently, the effective number of degrees of freedom N* will often be much less than the number of years N, so that 95% confidence intervals will be broader than those assumed in the submitted paper, and some trends (Tables 1 and S1) that are considered statistically different from zero at the 0.05 level in the paper probably are not.

Given that trend estimation and determination of their confidence intervals is a key component of this paper, I consider a major revision of the paper will be required to take serial correlation (autocorrelation) of time series into account.

MAJOR COMMENTS

The "Data Processing" section of the paper is extremely succinct about the methods used for the computation of trends (p. 9, lines 3-6), the computation of correlation coefficients (p.10, lines 1-3) and their confidence intervals and significance levels. Upon examining the trends and correlation coefficients considered statistically different from zero in the text and Tables, I came to the conclusion that the authors used statistical methods that assume iid (independent identically distributed) variables. This is a major issue, because neglecting autocorrelation (aka serial correlation) in the time series can invalidate the levels of statistical significance and confidence intervals that are found

throughout the text and in the Tables of results (von Storch and Zwiers 1999).

The authors did not perform a runs test for randomness to verify the underlying assumption that successive yearly values in their time series are independent of each other. Using the links provided by the authors, I downloaded yearly values (1950-2017) for the PDO index, and I downloaded monthly values for the NPGO index from which I then computed yearly averaged values from 1950 to 2017. Using a runs test for randomness on the PDO time series, the null hypothesis that the yearly values of the PDO index come in random order is rejected (p = 2.5e-08). Likewise, using a runs test for randomness on the NPGO time series, the null hypothesis that the yearly values of the NPGO index come in random order is also rejected (p = 0.0053).

Given this, it appears that neglecting to take into account autocorrelation in the statistical analyses of trends (for individual time series) and correlation coefficients (for paired time series) is an important flaw in a paper whose intent is to focus on the influence of decadal climate variability on yearly oxygen and nutrient time series. But luckily this flaw can be fixed, as methods that account for autocorrelation by adjusting the number of effective degrees of freedom do exist.

For trends calculations, Thomson and Emery (2014) propose dividing the length of the time series N by an integral time scale T in order to obtain an effective number of degrees of freedom $N^* = N/T$. Using their method for estimating an integral time scale by including lags of up to plus or minus 10 years for example, I obtained an integral time scale for the PDO time series of 2.8 years. This implies that on average, we get independent values of the PDO index every 3 years or so.

Autocorrelation of time series is not only a problem for trends estimation. It also affects confidence intervals for correlation coefficients of paired time series. When we have N pairs of (x,y) values, the effective number of degrees of freedom $N^*$ is not equal to N-2 as assumed by the authors, but is generally smaller. A resampling scheme that reduces the impact of serial correlation on inferences made about the correlation coef-

ficient is described by Ebisuzaki (1997), who also provides a review of other techniques that correct for autocorrelation in the estimation of confidence intervals for correlation coefficients.

MINOR COMMENTS

1. p.6, line3-4; and 5 CONSECUTIVE months of at least -0.5°C are defined as La Niña events

2. p.6, line 16; the discarding OF already sparse data

3. p.8, lines 14-16; the squares and circles are difficult to tell apart from each other, possibly because they are too small. Using circles and triangles (instead of squares) might provide a better visual contrast. This comment about squares and circles also applies to figures 3, 4, 5, 7, S2, S3, S4, S5.

4. p.9, line 22; Do the spatial scales of 50 m and 100 m apply to both the horizontal and vertical directions? If not, the word "spatial" should be replaced with either vertical or horizontal.

5. p.15, line 20; replace thEn with thAn

6. p.17, lines 25-27; A correlation is significant at r = -0.27, and another is not significant at r = 0.25. But one suspects that the p-value associated with r = -0.27 is barely below 0.05 whereas the p-value associated with r = 0.25 is barely above 0.05. A growing number of scientists insist that giving the actual p-values in the text and in tables is actually preferable to the dichotomy of dividing them in two categories: significant or not significant. See McShane et al. (2019) and other papers in a special issue of The American Statistician entirely dedicated to "A world beyond p < 0.05". I would love to see more p-values throughout the text and tables of this paper.

7. p.29, line 4; "Trends not within the 95% confidence interval are shown in italics" probably does not convey the intended message. Based on the table, I am guessing the intent was to write "Trends whose 95% confidence interval includes zero are shown

in italics", or something like that.

8. p.31; Table 2; Add 2 columns with p-values, one for the PDO-correlation, and the other for the NPGO-correlation. The effective number of degrees of freedom N*-2 (not N-2) should be used to estimate p-values.

9. Figure 6; add the label "Lag (years)" to the the x-axis of the left and right panels.

REFERENCES

Ebisuzaki, W., 1997. A method to estimate the statistical significance of a correlation when the data are serially correlated. Journal of Climate, 10(9), 2147-2153.

McShane, B.B., Gal, D., Gelman, A., Robert, C. & Tackett, J.L., 2019. Abandon Statistical Significance. The American Statistician, 73:sup1, 235-245.

Von Storch, H. & Zwiers, F. W., 1999. Statistical analysis in climate research. Cambridge University Press, 494p.

Thomson, R. E., & Emery, W. J., 2014. Data analysis methods in physical oceanography. Newnes, 716p. See section 3.15.2 on "Trend estimates and the integral timescale".

---

## Referee Comment (RC3) · Anonymous Referee #3 · 27 Aug 2019

What is new in this manuscript (ms)? It is difficult to determine how much is original.

The authors have completed an excellent synthesis of many data sources over many decades. This manuscript covers changes in four ocean properties in seven ocean areas and compares these changes to five climate indices and the global ocean surface temperature, providing more than one hundred possible relationships to be presented and evaluated.

I ask the authors to examine years with sparse data to determine how representative and accurate they might be. For example, if oxygen and nutrients vary in the opposite direction, does this relationship hold in years of sparse data? If changes in nutrients are in the same direction, then does this relationship hold in years of sparse data?

[Figure]

Why not plot the standard error of the mean values of oxygen and nutrients for each year? These standard errors might provide visual insight into the impact of years of sparse data on the trends and correlations.

Data deemed unacceptable by Schmidko et al. (2017) are included in this ms to avoid discarding already sparse data. The ms notes possible errors in density that might arise from including these data but does not give expected errors in oxygen or nutrients. Nor do they explain why errors in density are relevant to this ms.

In calculating correlations among time series, how are the number of degrees of freedom determined? Convince the readers that the number of degrees of freedom are determined appropriately.

I prefer that the title state that the manuscript gives results for "... the depth range of 50 to 300 metres in selected areas of the North and equatorial Pacific". As for "influence of decadal oscillations on . . . trends", the trends over the full time series as shown in figures 3, 4, 5, and 7 do not take into account the impact of PDO and NPGO decadal signals on these trends from the 1950s to recent years. I believe the variability in the oxygen and nutrient time series related to PDO, NPGO and other climate decadal oscillations should be removed from the data before 50 to 70-year trend is determined. Such an adjustment would allow the manuscript to match its title.

Why are graphs for Area P, Peru and Aloha in the Supplement, whereas graphs for other areas are in the ms? Please put them all together in the ms.

The years of two maxima and two minima seem to be close for the 18-year oscillation and the NPI. How correlated are these two series? Can their impacts be separated?

The names of all agencies that provide data and time series of indices need to be given, rather than only their Internet sites.

The writing in many places is sloppy and sometimes wrong. Too much information is included that clutters an already complex ms. I have given a few examples of these

features below, but all authors need to read all the ms carefully to deal with this issue.

Here are examples of sloppy and sometimes incorrect writing.

The manuscript states on page 2, last paragraph, that increases in ocean surface temperature influence oxygen concentration through changes in solubility of oxygen and changes in convection of oxygen to subsurface layers. This sounds reasonable to me. However, the sentence beginning on page 3, line 16, attributes oxygen changes to solubility changes only. This attribution is then contradicted in the following sentences.

On page 3 line 20, the ms notes that shoaling thermoclines during La Niña or cool (negative) PDO in the eastern Pacific enhance nutrient supply. Should this region be stated more accurately as eastern tropical Pacific? I expect there are regions of the eastern Pacific outside of the eastern tropical Pacific that behave otherwise during La Niña and negative PDO.

I was surprised by the definition of PDO as given on page 4 lines 21 to 23. It has been taken incorrectly from Dressler et al. (2010).

I believe the correct definition of El Nino and La Nina is "five consecutive 3-month periods .." (page 6, line 3).

On page 11, line 14, the sentence reads ". . .the linear trend of the oxygen content of the layer 50 to 300 m decreases for the entire time period . . ." Actually, the linear trend is constant and negative. The trend would not be linear if it decreased.

I doubt that Station P was occupied continuously from 1943 and it was likely established as a weather observation site rather than an ocean measurement site. (page 7, line 13)

Solid lines in Figures 3, 5, etc, are described in the captions as representing positive PDO phase after 1977, despite the obvious negative phase from 1998 to 2012.

Insert names of areas into the graphs of Figures 3 and 4. Give the units of ocean

properties in Figures 4, 5, and 7, as well as the units of trends. Lines in gray on figures will be more visible if black is used.

Examples of too much information:

I prefer that the Abstract begin with "Oxygen and nutrient time series since the 1950s were investigated at 50 to 300 metres depth in seven areas of the North and equatorial Pacific .." The sentences preceding this one in the present Abstract are not necessary and divert the reader from the essential content of this manuscript.

The paragraph on page 11 from lines 11 to 18 notes the many areas in which the linear trend decreases for the entire time period. (I assume the trend is negative rather than decreasing). However, the final sentence notes that oxygen trends are not significant for the entire time period except in two areas. Why describe insignificant trends at all? There are sufficient significant trends to provide enough information to overwhelm most readers.

The first 9 lines of page 12 describe numerical differences between this ms and previous studies. However, the depth ranges are different, and the years are different in the two studies. The information is not useful unless the differences are attributed to the depth range or years. This paragraph could be eliminated.

Regarding the subtropical convergence cell (STC), on page 16, lines 10-13, the authors note that, "Due to the long duration of the STC phases and the sparse data set, it is not possible to perform a meaningful correlation analysis to investigate STC influence on the oxygen and nutrient variations." In addition, the authors note on page 20, line 20, that the STC showed no clear signal in the equatorial Pacific. Given this lack of impact, why devote any text to the STC at all, except to say it does not have significant correlation with oxygen and nutrient time series, despite an expectation that it might?

---

## Author Comment (AC1) · 19 Sep 2019

Reply to reviewer 1

(our response **in bold font**)

**We thank the reviewer for the helpful comments. Actually, for trends computations the effective degrees of freedom of N\*=N/T was already used in our matlab routine for the submitted version, however we did not describe it in the text. We edited the text accordingly and now describe in detail how the degrees of freedom are derived. The autocorrelation of each time series is used to determine the statistical independence of measurements in successive years and the half folding width is used to determine the degrees of freedom of the timeseries. The routine used is included at the end of this text.**

GENERAL ASSESSMENT

Dr Lothar Stramma and coauthors compiled a mostly public dataset of oxygen, nitrate, silicate and phosphate observations from eight areas of the Pacific Ocean. Using those data, they calculated long-term trends over the entire length of the time series. They also calculated trends over the 1950 to 1976 negative phase of the Pacific Decadal Oscillation (PDO), and after 1976 for the positive phase of the PDO. They showed that in some cases, trend estimates were very different depending on sign of PDO phase.

In addition, they calculated lagged correlation coefficients between oxygen, nutrient and temperature time series from their eight areas with the PDO and with the North Pacific Gyre Oscillation (NPGO) climate indices. They also briefly examined how three other modes of climate variability (ENSO, STC, NPI) may or may not affect the time series in the same eight areas.

This is a worthwhile study. It is important that we better understand "The influence of decadal oscillations on the oxygen and nutrient trends in the Pacific Ocean". But there is a big caveat: the word DECADAL in the paper's title aptly captures the idea that successive years are not statistically independent of each other. For instance, when a given year is above normal for the PDO, the odds of having a below normal value of the PDO the following year is actually lower than 0.5; the dice are rigged. Several consecutive years with above normal values tend to be lumped together, and likewise for below normal values. From the point of view of statistical analysis, this implies the assumption that successive annual values in the oxygen, nutrients and temperature time series are independent of each other is not valid. Consequently, the effective number of degrees of freedom N\* will often be much less than the number of years N, so that 95% confidence intervals will be broader than those assumed in the submitted paper, and some trends (Tables 1 and S1) that are considered statistically different from zero at the 0.05 level in the paper probably are not.

Given that trend estimation and determination of their confidence intervals is a key component of this paper, I consider a major revision of the paper will be required to take serial correlation (autocorrelation) of time series into account.

**As explained above we had already used the effective degrees of freedom and mention it now more clearly in the text and we changed the serial correlation, as described also below. For the area P e.g. the degrees of freedom for oxygen trends were 13.8 for 23 available years for the period 1954- 1976, 33.9 for 40 years 1977 to 2017 and 50.8 for 63 years 1954-2017. The degrees of freedom depend strongly on the data set and time series analyzed. The method is now described in the text.**

MAJOR COMMENTS

The "Data Processing" section of the paper is extremely succinct about the methods used for the computation of trends (p. 9, lines 3-6), the computation of correlation coefficients (p.10, lines 1-3) and their confidence intervals and significance levels. Upon examining the trends and correlation coefficients considered statistically different from zero in the text and Tables, I came to the conclusion that the authors used statistical methods that assume iid (independent identically distributed) variables. This is a major issue, because neglecting autocorrelation (aka serial correlation) in the time series can invalidate the levels of statistical significance and confidence intervals that are found throughout the text and in the Tables of results (von Storch and Zwiers 1999).

**The Data processing explanation for the trend was extended with the information on the use of effective number of degrees of freedom for the computation of the confidence intervals, as well as the p-values for the correlations.**

The authors did not perform a runs test for randomness to verify the underlying assumption that successive yearly values in their time series are independent of each other. Using the links provided by the authors, I downloaded yearly values (1950- 2017) for the PDO index, and I downloaded monthly values for the NPGO index from which I then computed yearly averaged values from 1950 to 2017. Using a runs test for randomness on the PDO time series, the null hypothesis that the yearly values of the PDO index come in random order is rejected ($p = 2.5e-08$). Likewise, using a runs test for randomness on the NPGO time series, the null hypothesis that the yearly values of the NPGO index come in random order is also rejected ($p = 0.0053$). Given this, it appears that neglecting to take into account autocorrelation in the statistical analyses of trends (for individual time series) and correlation coefficients (for paired time series) is an important flaw in a paper whose intent is to focus on the influence of decadal climate variability on yearly oxygen and nutrient time series. But luckily this flaw can be fixed, as methods that account for autocorrelation by adjusting the number of effective degrees of freedom do exist. For trends calculations, Thomson and Emery (2014) propose dividing the length of the time series N by an integral time scale T in order to obtain an effective number of degrees of freedom $N^* = N/T$. Using their method for estimating an integral time scale by including lags of up to plus or minus 10 years for example, I obtained an integral time scale for the PDO time series of 2.8 years. This implies that on average, we get independent values of the PDO index every 3 years or so. Autocorrelation of time series is not only a problem for trends estimation. It also affects confidence intervals for correlation coefficients of paired time series. When we have N pairs of (x,y) values, the effective number of

degrees of freedom N* is not equal to N-2 as assumed by the authors, but is generally smaller. A resampling scheme that reduces the impact of serial correlation on inferences made about the correlation coefficient is described by Ebisuzaki (1997), who also provides a review of other techniques that correct for autocorrelation in the estimation of confidence intervals for correlation coefficients.

**As mentioned above we made the trend computations using the effective degrees of freedom N\*=N/T, however it was not described in the text and now a statement is added in the revised version. Successive years are not assumed to be statistically independent except the autocorrelation function does indicate so. Never the less, with short and non continuous time series, methods and statistical analysis is limited to a degree.**

MINOR COMMENTS

1. p.6, line3-4; and 5 CONSECUTIVE months of at least -0.5_C are defined as La Niña Events

**'CONSECUTIVE' added**

2. p.6, line 16; the discarding OF already sparse data

**'OF' added**

3. p.8, lines 14-16; the squares and circles are difficult to tell apart from each other, possibly because they are too small. Using circles and triangles (instead of squares) might provide a better visual contrast. This comment about squares and circles also applies to figures 3, 4, 5, 7, S2, S3, S4, S5.

**Triangles were also difficult to separate from circles and did not cover the entire crosses, therefore me modified all figures by showing the circles in magenta and the squares in blue.**

4. p.9, line 22; Do the spatial scales of 50 m and 100 m apply to both the horizontal and vertical directions? If not, the word "spatial" should be replaced with either vertical or horizontal.

**"spatial" was replaced by "vertical"**

5. p.15, line 20; replace thEn with thAn

**thEn replaced by thAn**

6. p.17, lines 25-27; A correlation is significant at r = -0.27, and another is not significant at r = 0.25. But one suspects that the p-value associated with r = -0.27 is barely below 0.05 whereas the p-value associated with r = 0.25 is barely above 0.05. A growing number of scientists insist that giving the actual p-values in the text and in tables is actually preferable to the dichotomy of dividing them in two categories: significant or not significant. See McShane et al. (2019) and other papers in a special issue of The American Statistician entirely dedicated to "A world beyond p < 0.05". I would love to see more p-values throughout the text and tables of this paper.

**The p-values were added to table 2 and statistical significance based on p-values were removed in the text. The references to the papers by McShane et al. 2019 and Amrhein et al. 2019 are referenced to explain why no significance statement is given.**

7. p.29, line 4; "Trends not within the 95% confidence interval are shown in italics" probably does not convey the intended message. Based on the table, I am guessing the intent was to write "Trends whose 95% confidence interval includes zero are shown in italics", or something like that.

**As proposed we modified the text in Table 1 and Table S1 to "Trends whose 95% confidence interval includes zero are shown in italics"**

8. p.31; Table 2; Add 2 columns with p-values, one for the PDO-correlation, and the other for the NPGO-correlation. The effective number of degrees of freedom N*-2 (not N-2) should be used to estimate p-values.

**p-values are now added in table 2.**

9. Figure 6; add the label "Lag (years)" to the the x-axis of the left and right panels.

**"Lag (year)"was added to both frames in Figure 6.**

REFERENCES

Ebisuzaki, W., 1997. A method to estimate the statistical significance of a correlation

when the data are serially correlated. Journal of Climate, 10(9), 2147-2153.

McShane, B.B., Gal, D., Gelman, A., Robert, C. & Tackett, J.L., 2019. Abandon Statistical

Significance. The American Statistician, 73:sup1, 235-245.

Von Storch, H. & Zwiers, F. W., 1999. Statistical analysis in climate research. Cambridge

University Press, 494p.

Thomson, R. E., & Emery, W. J., 2014. Data analysis methods in physical oceanography.

Newnes, 716p. See section 3.15.2 on "Trend estimates and the integral

timescale".

**The reference McShane et al. 2019 and in addition Amrhein et al. 2019 were included.**

**Matlab routine used for trend computations:**

```
% version for oxygen in micromoles/kg
q=load('wiwi2.dat');
t=q(:,1)'; tm=mean(t); t=t-tm;
y=q(:,2)';
E=[t',ones(size(t))'];
covmat=inv(E'*E);
model=covmat*E'*y';
res=E*model-y';
c_res=xcorr(res,'coeff');
ii=ceil(length(c_res)/2):length(c_res);
tscale=2*max(cumtrapz([0:length(ii)-1],c_res(ii)));
% next line introduced 3 May 2010
tscale = max([tscale,1]);
degfree=(size(E,1)/tscale-size(E,2));
chisqr=sum(res.^2)/degfree;
model_err=sqrt(diag(covmat)*chisqr);
```

```
y_model=t*model(1)+model(2);
y_model_err_95=student(degfree)*sqrt(model_err(2)^2+(t*model_err(1)).^2);
plot(t+tm,y,'r.',t+tm,y_model,'b-',t+tm,y_model+y_model_err_95,'m--',t+tm,y_model-y_model_err_95,'m--')
disp(['The linear trend is ',num2str(model(1),3),' micromoles/kg/yr'])
disp(['The formal 95% confidence intervals for that trend are +/- ',num2str(model_err(1)*student(degfree),3),' micromoles/kg/yr'])
% next line introduced 30 April 2010
dof=(size(E,1)-size(E,2))/tscale
```

---

## Author Comment (AC2) · 19 Sep 2019

Reply to reviewer 2

(our response **in bold font**)

**We thank the reviewer for the helpful comments.**

First, I am a physical oceanographer with insufficient knowledge on biogeochemistry, so I am not confident if I can properly judge this manuscript.

Biogeochemical measurements are definitely more difficult than physical ones, and the time series in each region shown in the manuscript must be valuable themselves. Nevertheless, Secs. 3.2 through 3.4, which should be the main result of this manuscript (inferred from its title), shows rough analyses with little plausible physical mechanisms.

**Right, biogeochemical measurements are more difficult than physical ones, and also less biogeochemical measurements are available. Therefore, very few observations of long-term nutrient changes exist, which was the motivation for this manuscript. Changes are observed to be related to long-term trends and in addition to different climate signals related physical mechanisms but also to local biological conditions. The inclusion of the proposed 137°E section helped to put the observations better into the context of the PDO.**

My biggest question is, in Sec. 3.2, why the authors show a linear trend for the whole period after 1976 (Fig. 3 and other figures) although they state that "the period 1998 to 2013 is dominated by negative seasonal mean PDO indices and is typically considered as a cool (negative) PDO phase" (Page 5, Line 3-5). If they are to see the relation between the biogeochemical variability and PDO, don't they need to calculate the trend for each of three periods (-1976, 1977-1998, and 1998-2013)?

**As the data base for nutrient measurements is small compared to oxygen and temperature measurements especially in regions with no continuous measurements, we think that another subdivision would stress the data set too much. As written in the text in the areas E and D the nutrient data base is so low, that we even did not show the nutrient trend figures. For the area 2-5°S 84-87°W (Fig. 4) only two measurements are available for the period 1998 to 2013 and in the Peru region (now Suppl. Fig. S6) there is a data gap between 1985 and 2008. We added in the concluding results: "…the results might have larger uncertainties for the areas with low data coverage and the combination of the warm and cold PDO periods after 1976". While the reviewer is correct that the overall trend here is not that meaningful with the underlying strong variability, never-the-less it is presented for constancy with the other areas.**

Furthermore, although "it is expected that during cold PDO phases the oxygen will decrease and the nutrients increase in the eastern equatorial and tropical Pacific, while during warm PDO periods the oxygen should increase and the nutrients decrease" (Page 13, Line 11-13), the observed trends in areas E, D, G were opposite. So, what is the mechanism? As a non-expert in this field, I feel a bit hard to find what the new findings of this manuscript are.

**The expectation mentioned on page 13 lines 11-13 is based on a possible PDO influence on the thermocline depth in a model by Deutsch et al., 2011 and a general Pacific Ocean description by Chavez et al., 2003. The new finding is that in real measurements these changes can't be always seen, which means that other mechanisms are influencing the oxygen and nutrient distribution and local changes have to be validated by measurements.**

Other comments:
Sec. 2.1: Subtropical cell (STC) is an ocean circulation component and is not temporospatial variability. Therefore, I feel odd to see that STC is aligned with climate variability such as PDO, NPGO, and ENSO as a controlling factor.

**As mentioned in the text, according to Hong et al., 2014 the STC is strongly associated with the PDO. However, as model simulations by Duteil et al. 2014 described changes in oxygen and phosphate transport, we wanted to check this with measurements.**
**Still, the reviewer is correct, STCs can be modified and rely on the PDO, we modified the manuscript and now excluded STCs from our analysis, to focus more on the trends with significant impact.**

Sec. 2.2: The authors' data do not cover the western part of the North Pacific Ocean (Fig. 2). Why not the authors use the 137E repeat hydrographic section maintained by the Japan Meteorological Agency since 1967 although one of them belongs to the agency? With high temporal resolution and large spatial (meridional) extent, the section is expected greatly to fill the data gaps.

**Now an area of the 137°E section is included to better cover also the Northwestern Pacific. The added area helped a lot to describe the results of the different areas in this manuscript in relation to the PDO.**

Page 14, Line 19-20, "probably caused by water masses propagating by 5 to 15 years from Oyashio region into this part of the North Pacific": why do the authors consider horizontal advection for the area P only?

**This water would propagate further southeastward with the subtropical gyre towards the CalCOFIc region. The other regions in the North Pacific show a larger correlation with the PDO and this is now mentioned in the text. Of course water mass propagation might influence all areas, and this is mentioned now in the concluding remarks.**

Secs. 3.2-3.4: If the authors are to extract decadal variability superimposed on the long-term trend (Sec. 3.1), it is better to examine the time series after subtracting the long-term trend.

**Reviewer 3 proposed to go the opposite direction, remove first the PDO, NPGO and other climate trends before computing the long-term trend. The long-term trends might not be only related to ocean warming but also the PDO and other climate signals. Hence removing the long-term trend first might remove also the contribution by PDO and other signals, therefore we did not remove the long-term trend first and computed the PDO signal related to the observed oxygen and nutrient changes. For a time series of significant lengths, with several oscillations of the overlying signal this certainly would be the best approach, but since the data time series is short, any long term trend certainly is influenced by the phase of the oscillation at the beginning and end of time series, thus making this approach less ideal.**

---

## Author Comment (AC3) · 19 Sep 2019

Reply to reviewer 3

(our response **in bold font**)

**We thank the reviewer for the helpful comments.**

What is new in this manuscript (ms)? It is difficult to determine how much is original.
The authors have completed an excellent synthesis of many data sources over many
decades. This manuscript covers changes in four ocean properties in seven ocean
areas and compares these changes to five climate indices and the global ocean surface
temperature, providing more than one hundred possible relationships to be presented
and evaluated.

I ask the authors to examine years with sparse data to determine how representative
and accurate they might be. For example, if oxygen and nutrients vary in the opposite
direction, does this relationship hold in years of sparse data? If changes in nutrients
are in the same direction, then does this relationship hold in years of sparse data?

**We are not sure if we understand the request of the reviewer here. Intra annual or short term
variations in oxygen and nutrients can have a very different origin than long term variability,
and they are not necessarily connected. Such an analysis, while interesting, would be
possible for areas with sufficient data like CalCOFI, though we think the focus of the present
manuscript should be on longer term changes. As the reviewer said, with already many
options we decided to exclude the STCs to focus more on larger-scale correlations.**

Why not plot the standard error of the mean values of oxygen and nutrients for each
year? These standard errors might provide visual insight into the impact of years of
sparse data on the trends and correlations.

**Despite a simple task at glance, the data processing and in particular gathering was
programmed in a way to bin data into annul data point early in the progress during data
aquisition. To accommodate this desirable extension, we would have to change the
computation of all areas used. Due to the often low amount of available data we combined all
available data from one year into one profile which leads to a different amount of data points
at different depth layers at different years and the error bars would not be comparable. A
better option would be to no bin the data and work with the raw data, applying statistics that
can handle highly heterogeneous data distributions, though this would make the data
handling significant more complicated to follow, with no gain for the outcome and results. For**

**more localized studies this would probably be justified, putting more emphasize on the study of regional variability.**

Data deemed unacceptable by Schmidko et al. (2017) are included in this ms to avoid discarding already sparse data. The ms notes possible errors in density that might arise from including these data but does not give expected errors in oxygen or nutrients. Nor do they explain why errors in density are relevant to this ms.

**We rewrote the text to describe in more detail why this was done and is justified from our point of view.  The text now reads:**

Quality control and handling is described in Schmidtko et al. (2017) for oxygen and used here similarly for nutrients. The only divergence to the described procedure was that  bottle data with missing temperature and/or salinity were assigned the temporal and spatial interpolated temperature and salinity derived from MIMOC (Schmidtko et al., 2013).  This was done to ensure all data were in $\mu mol\ kg^{-1}$ and not requiring the discarding of already sparse data. In Schmidtko et al. (2017) this was not performed, since the error introduced near or in boundary currents and fronts can be significant. In contrast the areas here are chosen to represent homogeneous patches with significant amount of data in the open ocean, thus in the areas analyzed here, this may only lead to minor errors in density resulting in an error of less than 0.05%, therefore negligible small in  $\mu mol\ kg^{-1}$, compared to the oxygen or nutrient data accuracy

In calculating correlations among time series, how are the number of degrees of freedom determined? Convince the readers that the number of degrees of freedom are determined appropriately.

**As requested by reviewer 1 we modified the text explaining how the effective degrees of freedom were derived.**

I prefer that the title state that the manuscript gives results for "... the depth range of 50 to 300 metres in selected areas of the North and equatorial Pacific". As for "influence of decadal oscillations on : : : trends", the trends over the full time series as shown in figures 3, 4, 5, and 7 do not take into account the impact of PDO and NPGO decadal signals on these trends from the 1950s to recent years. I believe the variability in the oxygen and nutrient time series related to PDO, NPGO and other climate decadal oscillations should be removed from the data before 50 to 70-year trend is determined. Such an adjustment would allow the manuscript to match its title.

**Reviewer 2 proposed to go the opposite direction, remove first the long-term trend before computing the PDO and NPGO trends. The PDO and NPGO trends might contribute to the long-term trend. Hence removing the PDO and NPGO trends first might also remove the contribution to the long-term trend, therefore we did not remove the PDO and NPGO trends first and computed the different trends from the original data set.  We changed the title to now state the depth interval from 50-300 metres.**

Why are graphs for Area P, Peru and Aloha in the Supplement, whereas graphs for

other areas are in the ms? Please put them all together in the ms.

**The figures all look similar, hence showing all figures, including now an additional area proposed by reviewer 2, in the main text would be redundant as the main information is in Table 1. For those interested in details of all areas can reference the supplement.**

The years of two maxima and two minima seem to be close for the 18-year oscillation and the NPI. How correlated are these two series? Can their impacts be separated?

**Different to the earlier version, we now use the same time period of the oxygen data sets which led to different results. The maxima and minima are close and cannot always be separated depending on data coverage. However, the correlation of the 18.6 year oscillation is much weaker than the NPI and we think it is an interesting result that these two oscillations lead to quite different correlations, reasons for this can bethe shorter-time fluctuations in the NPI or the sensitivity to phase shifts in the oscillation. This is now stated in the text.**

The names of all agencies that provide data and time series of indices need to be given, rather than only their Internet sites.

**'The names of the agencies/universities are included in the paragraph data availability with the web-pages'**

The writing in many places is sloppy and sometimes wrong. Too much information is included that clutters an already complex ms. I have given a few examples of these features below, but all authors need to read all the ms carefully to deal with this issue. Here are examples of sloppy and sometimes incorrect writing. The manuscript states on page 2, last paragraph, that increases in ocean surface temperature influence oxygen concentration through changes in solubility of oxygen and changes in convection of oxygen to subsurface layers. This sounds reasonable to me. However, the sentence beginning on page 3, line 16, attributes oxygen changes to solubility changes only. This attribution is then contradicted in the following sentences.

**As we mentioned convection on page 3, we thought we do not need to list all components again on page 3. Now we included on page 3: '…changes in convection and thermocline depth'. We also removed STC analysis from the manuscript to focus more on the indices with larger impacts.**

On page 3 line 20, the ms notes that shoaling thermoclines during La Niña or cool (negative) PDO in the eastern Pacific enhance nutrient supply. Should this region be stated more accurately as eastern tropical Pacific? I expect there are regions of the eastern Pacific outside of the eastern tropical Pacific that behave otherwise during La

Niña and negative PDO.

**The eastern Pacific was mentioned at the beginning of the sentence, however to make it more clear that the related changes appear in the eastern Pacific we added ' in the eastern Pacific' in the sentence mentioned as well as in the next sentence.**

I was surprised by the definition of PDO as given on page 4 lines 21 to 23. It has been taken incorrectly from Dressler et al. (2010).

**We assume Dressler et al. (2010) should be Deser et al. (2010). According to the definition of Deser et al. (2010) one word was missing, which is included in the revised version.**

I believe the correct definition of El Nino and La Nina is "five consecutive 3-month periods .." (page 6, line 3).

**Various definitions of El Nino and La Nina do exist, the three months running mean was mentioned at the beginning of the sentence, however to make the definition more clear we modified the text as requested.**

On page 11, line 14, the sentence reads ": : :the linear trend of the oxygen content of the layer 50 to 300 m decreases for the entire time period : : :" Actually, the linear trend is constant and negative. The trend would not be linear if it decreased.

**Thanks for this information, 'decreases' is now replaced by 'is negative'**

I doubt that Station P was occupied continuously from 1943 and it was likely established as a weather observation site rather than an ocean measurement site. (page 7, line 13)

**The description of Station P was taken from Wikipedia and is an example that in short summaries in Wikipedia wrong information might be included. Thanks for letting us know. Now the text reads:**

**Station P, located at 50°N, 145°W in the North Pacific, was established as a weather observation site with a weather ship in 1949 which was manned continuously until 1981, and routinely hydrographic measurements were started in the 1950's. After the termination of the weather ship program shipboard measurements were made on average 3 times a year since.**

Solid lines in Figures 3, 5, etc, are described in the captions as representing positive PDO phase after 1977, despite the obvious negative phase from 1998 to 2012.

**As the data base for nutrient measurements is small compared to oxygen and temperature measurements especially in regions with no continuous measurements, we think that another subdivision would stress the data set too much. As written in the text in the areas E and D the**

**nutrient data base is so low, that we did not show the nutrient figures. For the area 2-5°S 84-87°W (Fig. 4) only two measurements are available for the period 1998 to 2013 and in the Peru region (now Suppl. Fig. S6) there is a data gap between 1985 and 2008. Hence we decided to define the entire period after 1977 as positive PDO in the text and the figure legends refer to the definition used for this manuscript.**

Insert names of areas into the graphs of Figures 3 and 4. Give the units of ocean
properties in Figures 4, 5, and 7, as well as the units of trends. Lines in gray on figures
will be more visible if black is used.

**The names Area E,D and G are now included in figures 3 and 4. Units for the parameters are given now in the figure legends of figures 4, 5, 7 as well as S3 to S6 as: (in µmol kg$^{-1}$, red crosses) while the units for the trends were already mentioned in the figure legends. Yes black lines for the PDO are more visible, but as the PDO curve is just background information, we think that the PDO as black line dominates the figure too much (see figure below).**

[Figure]

Examples of too much information:
I prefer that the Abstract begin with "Oxygen and nutrient time series since the 1950s
were investigated at 50 to 300 metres depth in seven areas of the North and equatorial
Pacific .." The sentences preceding this one in the present Abstract are not necessary
and divert the reader from the essential content of this manuscript.

**According to the Geosciences guidelines for authors for the abstract: 'After a brief introduction to the topic, the summary recapitulates the key points of the article and mention possible directions for prospective research.' We prefer to start with an introduction instead of starting right away with the results.**

The paragraph on page 11 from lines 11 to 18 notes the many areas in which the linear
trend decreases for the entire time period. (I assume the trend is negative rather than
decreasing). However, the final sentence notes that oxygen trends are not significant
for the entire time period except in two areas. Why describe insignificant trends at all?
There are sufficient significant trends to provide enough information to overwhelm most

readers.

**As requested by reviewer 1 significance should be avoided and the decision of good or bad should not be related to one fixed value. Hence we mention now only that areas are within or out of the 95% confidence interval.**

The first 9 lines of page 12 describe numerical differences between this ms and previous studies. However, the depth ranges are different, and the years are different in the two studies. The information is not useful unless the differences are attributed to the depth range or years. This paragraph could be eliminated.

**As earlier investigations in this region exist it seems reasonable to mention these results. Despite the different depth layers the comparison with earlier investigations indicates that changes are related to time periods analysed and not just a linear trend. This in now explained in more detail in the text. The bi-decadal trend is no longer mentioned in the respective paragraph as this is the subject of the later paragraph.**

Regarding the subtropical convergence cell (STC), on page 16, lines 10-13, the authors note that, "Due to the long duration of the STC phases and the sparse data set, it is not possible to perform a meaningful correlation analysis to investigate STC influence on the oxygen and nutrient variations." In addition, the authors note on page 20, line 20, that the STC showed no clear signal in the equatorial Pacific. Given this lack of impact, why devote any text to the STC at all, except to say it does not have significant correlation with oxygen and nutrient time series, despite an expectation that it might?

**The reviewer is correct; we now exclude the analysis of STCs in the manuscript to focus more on the indices with large-scale impact. This made the manuscript more focused and the reader is hopefully less distracted by indices that we do not find to correlate significantly.**

---

## Author Response (AR2)

**Associate Editor Decision: Publish subject to minor revisions (review by editor)** (05 Dec 2019) by Marilaure Grégoire

Comments to the Author:

Dear Lothar,

I have read the reviews provided by two reviewers of the revised version of your manuscript. One of the reviewers who assessed the first version of your work is satisfied by the answers to his/her comments and provides a few additional comments while another one is asking more justifications on the significance of the trend in O2 and nutrient data. I recommend that you address these last comments in a revised version.

Thank you so much for your efforts,

Kind regards,

Marilaure

**Suggestions for revision or reasons for rejection (will be published if the paper is accepted for final publication)**

It is difficult to have any confidence in the trend results. I would suggest writing a paragraph on the confidence in the trend results and their significance. Beyond purely statistical analysis, why are these trends significant?

*We added a paragraph dealing with the uncertainties of the mathematically significant trends. We argue that the observed trends indicate a relationship that needs to be monitored and revisited with future monitoring data, to understand better the causes of this relationship and the extent to which this is a related process to analyze in more detail in biogeochemical models and observational process analysis.*
*In the concluding remarks we now write concerning the PDO:*
*„Statistically significant trends and correlations hold true for the period and data analysis, though in data sparse regions, these findings should be subject of future scrutiny. Regional and small scale processes, related or independent of large scale PDO forcing may alter the signals seen so far.“*
*And further below addressing the other indices:*
*„With respect to other climate indices, the results are more mixed,* **and statistical significance should be treated with caution.** *“*

**Suggestions for revision or reasons for rejection (will be published if the paper is accepted for final publication)**

The definition of the Pacific Decadal Oscillation (PDO) is not correct. I have listed below the correct definition, as provided by Deser et al. (2010) on page 143, as well as the incorrect definition provided by Stramma et al. (this revised manuscript) on page 4 lines 21-23. I believe that Stramma et al. have missed the significance of the parentheses in Deser (2010).

Here is the description of the Pacific Decadal Oscillation as provided by Deser et al. (2010):
"The leading EOF of monthly SST anomalies over the North Pacific (after removing the global mean SST anomaly) and its associated PC time series are termed the Pacific Decadal Oscillation (PDO) after Mantua et al. (1997)."

Here is the description as provided by the authors of this manuscript:
"The PDO is the leading empirical orthogonal function (EOF) of monthly SST anomalies over the North Pacific between 20°N and 70°N, after removing the global mean SST anomaly and its associated principal component (PC) time series (Mantua et al., 1997; Deser et al., 2010; Newman et al., 2016)."

*Thank you for pointing this out. We used in all our computations the Deser et al. 2010 definition, though our description was formally not completely clear about the order of steps undertaken. We now use the identical sentence to Deser et al. as recommended by this reviewer but explain the abbreviations EOF and PC and reference this as citation.*

---

## Author Response (AR3)

**Associate Editor Decision: Publish subject to technical corrections** (16 Jan 2020) by Marilaure Grégoire

Comments to the Author:

Dear Lothar,

I have read the last version of your manuscript and I recommend its publication in Biogeosciences provided that you addressed a few technical corrections listed below.

Thank you for your efforts in addressing all the reviewers comments,

Kind regards,

Marilaure

*Thanks a lot for reading the manuscript and mention some technical corrections. We addressed the technical corrections and list the relevant changes below. In addition, we finalized the open references in the paragraph data availability.*

Minor technical comments:

Abstract line 4 "have" instead of "has"

*Changed accordingly*

Section 2.1, line 25: (after "removing the global mean" please remove anomaly )

*We modified the sentence as we did not compute the PDO ourselves but present here only the definition as written in Deser et al. 2010 where anomaly was written and hence show the original text of this publication. In the figures we plotted the PDO time series as downloaded from the web.*

Page 5, line 16, a bracket is missing NPI (..)

*Missing bracket added.*

Line 22: Please use SST throughout the manuscript for consistency

*We used SST where sea surface temperature is mentioned, however the GISTemp in Figure 1 is global land and sea surface temperature and we use for GISTemp 'global surface temperature'.*

Conclusion, line 8, "of data analysis"?

*Changed accordingly.*

Page 23, first line, "have also been described"

*Changed accordingly.*

Supplement material

Line 13: shown and not shows

*Changed accordingly.*

Legend Figure S2: first line, remove concentration for temperature and add red crosses for clarity

*'concentration' was removed and '(red crosses)' included in the figure legend.*

[revised manuscript text omitted]
 1982 and 1995 were removed as they were to high/low (1982: 236 µmol kg$^{-1}$, 1995: 185 µmol kg$^{-1}$), for nitrate 1959 was removed 14.4 µmol kg$^{-1}$ and for silicate 1983 23.7 µmol kg$^{-1}$. El Niño years defined as strong are marked by an additional magenta circle, strong La Niña years by an additional blue square. The change of the PDO status in 1977, 1999 and 2013 are marked by vertical dotted lines. PDO annual mean time series are shown in the oxygen time series and the NPGO annual mean time series in the nitrate, silicate and phosphate time series as solid grey lines.

[Figure]

**Figure S6.** Annual mean concentration (in µmol kg$^{-1}$, red crosses) for years available and trends for the layer 50 to 300 m plotted for the entire time period (dashed red lines) and for the periods 1950 to 1976 for the negative PDO phase and after 1976 for the positive PDO phase (solid red lines) between 7° and 12°S, 78 and 83°W from hydrodata CTD and bottle data in µmol kg$^{-1}$ yr$^{-1}$ for a) oxygen, b) nitrate, c) silicate and d) phosphate. For oxygen measurements in 1982 were removed as they were to high (62.4 µmol kg$^{-1}$). El Niño years defined as strong are marked by an additional magenta circle, strong La Niña years by an additional blue square. The change of the PDO status in 1977, 1999 and 2013 are marked by vertical dotted lines. PDO annual mean time series are shown in the oxygen time series and the NPGO annual mean time series in the nitrate, silicate and phosphate time series as solid grey lines.

Supplementary references:

Deser, C., Alexander, M. A., Xie, S.-P., and Phillips, A. S.: Sea surface temperature variability: Patterns and mechanisms, Annu. Rev. Mar. Sci., 2, https://doi.org/10.1146/annurev-marine-120408-151453, 2010.

Huang, B., Thorne, P. W., Banzon, V. F., Boyer, T., Cherupin, G., Lawrimore, J. H., Menne, M. J., Smith, T. M., Vose, R. S., and Zhang, H.-M.: Extended reconstructed sea surface temperature, version 5 (ERSSTv5): Upgrades, validations, and intercomparisons, J. Climate, 30, https://doi.org/10.1175/JCLI-D-16-0836.1, 2017.

5  Rayner, N. A., Parker, D. E., Horton, E. B., Folland, C. K., Alexander, L. V., Rowell, D. P., Kent, E. C., and Kaplan, A.: Global analyses of sea surface temperature, sea ice, and night marine air temperature since the late nineteenth century, J. Geophys. Res. 108 D14, 4407, https://doi.org/10.1029/2002JD002670, 2003.